# Nonparametric Variational Regularisation of Pretrained Transformers

**Fabio Fehr**[†‡] **& James Henderson**[†]
[†] Idiap Research Institute, Switzerland
[‡] Ecole Polytechnique Fédérale de Lausanne, Switzerland
{fabio.fehr,james.henderson}@idiap.ch

## Abstract

Pretrained transformers have demonstrated impressive abilities, but tend not to generalise well out-of-domain and are very expensive to fine-tune on new domain data. Nonparametric Variational Information Bottleneck (NVIB) has been proposed as a regulariser for training cross-attention in transformers, potentially addressing this domain overfitting problem. We extend the NVIB framework to replace all types of attention functions in transformers. We show that existing pretrained transformers can be reinterpreted as nonparametric variational models using an empirical prior distribution and identity initialisation with controllable hyperparameters. We then show that changing the initialisation introduces a novel, information-theoretic post-training regularisation in the attention mechanism, which improves out-of-domain generalisation on NLP tasks without any additional training. This success supports the hypothesis that the way pretrained transformer embeddings represent information is accurately characterised by nonparametric variational Bayesian models.

## 1 Introduction

Self-supervised pretraining of transformer models (Devlin et al., 2019; Lewis et al., 2020; Raffel et al., 2020; Zhang et al., 2022) has been hugely successful, improving performance in virtually every NLP task. This indicates that the inductive bias of attention-based representations is extremely effective for language, but it is not clear which characteristics of transformers are essential to this inductive bias and which are implementation details. We shed light on this issue by developing a variational Bayesian reinterpretation of pretrained transformers, which more explicitly characterises how information about the real data distribution is represented. This insight allows us to improve the models' out-of-distribution generalisation without any additional training.

In previous work, Henderson & Fehr (2023) derive a variational Bayesian generalisation of a single-head cross-attention layer of transformers, called Nonparametric Variational Information Bottleneck (NVIB). When used during training, the NVIB layer provides an information-theoretic, sparsity-inducing regulariser over attention-based representations. In this paper, we investigate the possibility that NVIB also provides an accurate theoretical model of existing attention-based models which have been trained without NVIB regularisation. We extend NVIB to all the uses of attention in transformers (single- and multi- head; cross- and self- attention; in encoders and decoders), and propose a method for converting a pretrained transformer into the weights of an equivalent nonparametric variational (NV) Bayesian model (NV-Transformer).[1]

This NV-Transformer is Bayesian in that it embeds text into a probability distribution over transformer embeddings, but remains equivalent by adding uncertainty around the embedding computed by the pretrained transformer, as illustrated in Figure 1. Theoretically, the exact equivalence only occurs when this uncertainty is exactly zero, but empirically

---

[1]https://github.com/idiap/nvr_transformers and https://github.com/idiap/nvib.

we find a practical range of non-zero uncertainty levels where the model's predictions are unchanged (Section 4.1). Continuing training of this initial model with NVIB regularisation would push this uncertainty higher, but training large models is computationally expensive and requires significant amounts of in-domain data. Instead, we only adjust hyperparameters of the initialisation to better satisfy the NVIB regulariser by increasing uncertainty, without any backward-passes or parameter updates.

Increasing the uncertainty level does change the model's predictions, but interestingly it does not degrade the model's task accuracy. Instead, it adds a form of post-training regularisation, which accesses a space of models which generate different outputs with the same level of accuracy (Section 4.1). Surprisingly, this post-training regularisation even improves out-of-domain generalisation (Section 4.2). These results show that adding smaller amounts of uncertainty removes unreliable information which does not generalise in-domain and can cause overfitting out-of-domain. This agreement between uncertainty in the Bayesian model and unreliability in pretrained transformer embeddings shows that our Bayesian reinterpretation helps characterise how transformers represent information.

**Contributions** In this work we contribute technically to the development of novel models which use Nonparametric Variational Information Bottleneck regularisation, and to the understanding of pretrained transformers. **1.** We extend NVIB regularisation beyond single-head, cross-attention to all forms of multi-head attention in a transformer, resulting in our proposed NV-Transformer architecture. **2.** We define a novel NVIB prior using the empirical distribution of a model's embeddings given a small amount of data (Section 3.2). **3.** We propose a reinterpretation of pretrained transformers as variational Bayesian models by defining a novel identity initialisation for NVIB with its controllable hyperparameters (Section 3.1). Adjusting these hyperparameters regularises the embeddings of this equivalent NV-Transformer without requiring retraining. **4.** Empirically, we show the usefulness of these proposals by smoothly varying the amount of this post-training regularisation (Section 4.1) and achieving improved performance in out-of-domain text summarisation and translation (Section 4.2). **5.** This successful Bayesian interpretation helps characterise the way that pretrained transformers represent reliable information in their embeddings (Section 5).

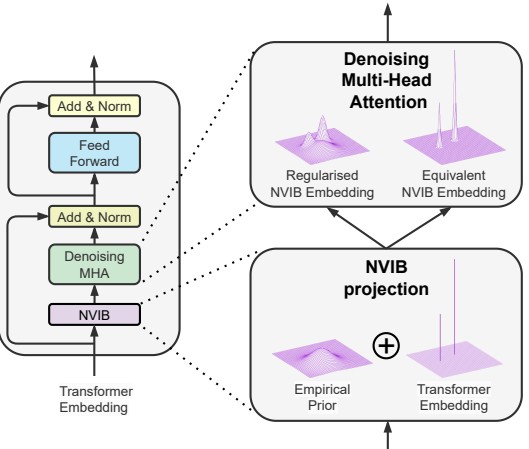

Figure 1: **Left:** Reinterpreted transformer encoder layer with NVIB and denoising multi-head attention (MHA). **Right:** A NVIB projection layer which combines the empirical prior with the transformer embeddings and produces a representation for denoising MHA for either post-training regularisation or exact equivalence.

## 2 Background

Henderson & Fehr (2023) propose NVIB as an information-theoretic regulariser for attention layers. It extends vector-space Variational Information Bottleneck (VIB) (Alemi et al., 2017) to the unboundedly large set-of-vectors representations supported by attention. Henderson & Fehr (2023) define a Variational Auto-Encoder (Kingma & Welling, 2014) for transformers with NVIB applied to its cross-attention layers. Behjati et al. (2023) apply NVIB to stacked transformer self-attention layers in the encoder, which when trained learns increasingly sparse representations across the layers of the model. They found that the stacked NVIB layers abstracted the character-level inputs into word-like units, when applied to the final layers of the encoder.

In variational Bayesian models like NVIB layers, the information which the input conveys about the latent representation is specified with a distribution over latent representations.

For attention-based representations, these distributions need to capture the essential structure imposed by the attention function on the sequence of vectors being accessed by attention, namely that these vectors are permutation invariant, have a normalised weighting, and can be arbitrarily many. A key insight of Henderson & Fehr (2023) is that these properties can be captured by interpreting attention-based representations as nonparametric mixture distributions. The accessed vectors are interpreted as specifying a mixture of impulse distributions (Equation (1) below). This is an equivalent representation in that we can define an exactly equivalent attention function in the form of Bayesian query denoising using this distribution as the prior. The query is interpreted as being corrupted by Gaussian noise, the attention weights are the query posterior's mixture weights given the query observation, and the output of the attention function is the expected value of this posterior distribution. The authors define a generalisation of attention known as *denoising attention* (Equation (2) below), which generalises the attention function to query denoising with any mixture distribution as the representation being accessed.

Since the number of vectors in a transformer embedding grows to accommodate the length of the input text, the latent space of equivalent mixture distributions is nonparametric in nature (i.e. there is no fixed set of parameters which can specify the entire space of mixture distributions). Thus, Henderson & Fehr (2023) define distributions over these generalised attention-based representations using Bayesian nonparametrics (Equation (3) below). In the remainder of this section we provide more details about denoising attention and NVIB's distributions over mixture distributions.

**Denoising attention** To show that the denoising attention function is a generalisation of the standard attention function, Henderson & Fehr (2023) provide a constructive proof of exact equivalence. The standard attention function projects the accessed set of vectors $Z \in \mathbb{R}^{n \times d}$ via weight matrices $W^K, W^V \in \mathbb{R}^{d \times d}$ to keys and values, respectively, and projects the accessing input vector $u' \in \mathbb{R}^{1 \times d}$ via the weight matrix $W^Q \in \mathbb{R}^{d \times d}$ to a query. It uses the keys' dimensionality $d$ for scaling. This scaled dot product attention function can be regrouped into a core dot product attention function $\text{Attn}(u, Z)$ in which all operations are done in the space of $Z$.

$$\text{Attention}(u', Z\ ;\ W^Q, W^K, W^V) = \text{Attn}(u' W^Q (W^K)^\top,\ Z)\ W^V = \text{Attn}(u, Z)\ W^V$$

where $u = (u' W^Q (W^K)^\top) \in \mathbb{R}^{1 \times d}$. The function $\text{Attn}(u, Z)$ can then be defined both in terms of a sum over the vectors $z_i$ in $Z$, or in terms of an integral over a distribution which is only non-zero at the $z_i$:

$$\text{Attn}(u, Z) = \text{softmax}\left(\tfrac{1}{\sqrt{d}} u Z^\top\right) Z\ =\ \text{DAttn}(u; F_Z)$$

$$F_Z = \sum_{i=1}^{n} \frac{\exp(\tfrac{1}{2\sqrt{d}}||z_i||^2)}{\sum_{i=1}^{n} \exp(\tfrac{1}{2\sqrt{d}}||z_i||^2)}\ \delta_{z_i} \tag{1}$$

$$\text{DAttn}(u;\ F) = \int_v \frac{f(v)\ g(u;\ v, \sqrt{d}I)}{\int_v f(v)\ g(u;\ v, \sqrt{d}I)\ dv}\ v\ dv \tag{2}$$

where $\delta_{z_i}$ is an impulse distribution at $z_i$, $f(\cdot)$ is the probability density function for distribution $F$, and $g(u;\ v, \sqrt{d}I)$ is the multivariate Gaussian function with diagonal variance of $\sqrt{d}$. $\text{DAttn}(u;\ F_Z)$ can be interpreted as Bayesian query denoising where $F_Z$ is the prior and $g(u;\ v, \sqrt{d}I)$ is a noisy observation of some true vector $v$. This construction shows that any transformer embedding $Z$ has an equivalent mixture of impulse distributions, namely $F_Z$, where denoising attention $\text{DAttn}(u;\ F_Z)$ gives us exactly the same result as attention $\text{Attn}(u, Z)$, for all queries $u$.

**Distributions over mixture distributions** Given the generalisation of attention-based representations to nonparametric mixture distributions, Henderson & Fehr (2023) propose to use Dirichlet Processes (DPs) to define distributions over this latent space of mixture distributions. Instead of directly computing the latent $Z$, their variational-Bayesian model computes a DP estimate of the posterior distribution over mixture distributions given the

input. More specifically, the model maps the input to a set of pseudo-observations, and then the posterior DP is computed with exact Bayesian inference given a prior DP and these observations. This posterior "$q$" distribution $DP(G_0^q, \alpha_0^q)$ consists of a base distribution $G_0^q$ for generating the vectors of the mixture distribution and a pseudo-count $\alpha_0^q$ for generating their mixture weights. The base distribution $G_0^q$ is a mixture with one Gaussian component for the prior "$p$" distribution $DP(\mathcal{N}(\boldsymbol{\mu}^p, \boldsymbol{I}(\sigma^p)^2), \alpha^p)$ plus one Gaussian component for each $i^{\text{th}}$ pseudo-observation $(\boldsymbol{\mu}_i^q, \sigma_i^q, \alpha_i^q)$ computed by the model:

$$G_0^q = \frac{\alpha^p}{\alpha_0^q} \mathcal{N}(\boldsymbol{\mu}^p, \boldsymbol{I}(\sigma^p)^2) + \sum_i \frac{\alpha_i^q}{\alpha_0^q} \mathcal{N}(\boldsymbol{\mu}_i^q, \boldsymbol{I}(\sigma_i^q)^2) \; ; \quad \alpha_0^q = \alpha^p + \sum_i \alpha_i^q \tag{3}$$

**Denoising attention during evaluation** As for other VAEs (Kingma & Welling, 2014), during evaluation Henderson & Fehr (2023) do not sample a mixture distribution $F$ from the posterior DP, but instead use the posterior's mean, which is its base distribution $G_0^q$. Denoising attention applied to $G_0^q$ can be computed straightforwardly because it is a mixture of Gaussians. This function differs from standard attention in that the component weights $\boldsymbol{\alpha}^q / \alpha_0^q$ replace the key biases implicit in the query-key dot product, and an interpolation between queries and values replaces the values in the attention-weighted average.

$$\text{DAttn}(\boldsymbol{u}; G_0^q) = \text{softmax}\left( \boldsymbol{u} \left( \frac{\boldsymbol{\mu}^q}{(\sigma^r)^2} \right)^{\top} + \underbrace{\boldsymbol{b}}_{\text{Bias}} \right) \left( \underbrace{\frac{(\sigma^q)^2}{(\sigma^r)^2} \odot (\mathbf{1}_n^{\top} \boldsymbol{u}) + \frac{\sqrt{d}}{(\sigma^r)^2} \odot \boldsymbol{\mu}^q}_{\text{Query-Value interpolation}} \right) \tag{4}$$

$$\boldsymbol{b} = \log(\frac{\boldsymbol{\alpha}^q}{\alpha_0^q}) - \left( \frac{1}{2} \left\| \frac{\boldsymbol{\mu}^q}{\sigma^r} \right\|^2 \right)^{\top} - \mathbf{1}_d \left( \log(\sigma^r) \right)^{\top} \tag{5}$$

where $(\sigma_i^r)^2 = (\sqrt{d} + (\sigma_i^q)^2)$, and $\mathbf{1}_d$ is a row vector of $d$ ones. When the attention bias term $\boldsymbol{b}$ is large across the non-prior components and thus down-weights the prior component, we can have an identical softmax to attention. When the posterior variance $(\sigma^q)^2$ is approximately zero, we get no interpolation between the query and value and get an identical value computation to attention. Henderson & Fehr (2023) propose this formulation of attention only for the case of single-head cross-attention during evaluation.

**Related Work** Related work in post-training regularisation such as quantisation (Dettmers et al., 2022; Yao et al., 2022; Xiao et al., 2023; Frantar et al., 2023b) and sparsity (Hubara et al., 2021; Frantar et al., 2023a; 2022) focus on regularising the model weights and not the latent embeddings. Similar to Frantar et al. (2023a), who propose a data-driven way to sparsify a model in one-shot without any retraining, we propose a data-driven form of soft sparsity for attention, without any retraining. Moreover, our method's regularisation uses information theory to regularise post-training. Our work shares similarity to the out-of-distribution generalisation literature in learning variational models with invariant features (Ilse et al., 2020) and parameter sharing in the embedding space to adapt to domain-shifts (Muandet et al., 2017; Li et al., 2017; Blanchard et al., 2021). We take a Bayesian approach and define our prior distribution based on empirical dataset statistics that allows for properties which are robust to domain shift. Park & Lee (2021) proposes a way to define pretrained language models as variational models by fine-tuning. Our work does not fine-tune or update the original model weights, nor does it train additional weights as in adapters (Houlsby et al., 2019; Hu et al., 2022). We keep the model weights frozen and map attention layers to NVIB layers, setting just a few hyperparameters based on validation performance.

## 3 A Nonparametric Variational Reinterpretation of Transformers

We propose a novel method for reinterpreting pretrained transformers as nonparametric variational Bayesian models. To support this construction, we implemented the following applications of NVIB: multihead denoising attention (Appendix B); encoder denoising self-attention (Appendix B.3); and decoder denoising causal attention (Appendix B.4).

Pseudocode for denoising attention is given in Appendix B.5. This allows us to apply NVIB to every form of attention used in standard transformers, resulting in our proposed NV-Transformer reinterpretation of pretrained transformers. Using these extensions, we propose an identity initialisation with controllable hyperparameters, which allows us to achieve an equivalence with the standard attention mechanisms in transformers (Section 3.1). We then define a novel empirical prior distribution, which introduces uncertainty in the latent representations in a way that captures the implicit uncertainty in pretrained transformers (Section 3.2).

## 3.1 Identity Initialisation

We define an identity initialisation for NVIB such that denoising attention is effectively equivalent to standard attention, inspired by Equation 1. Given a set of vectors $\mathbf{Z}$ input to the attention layer, our proposed NVIB layer converts it to the parameters $(\boldsymbol{\mu}, \boldsymbol{\sigma}, \boldsymbol{\alpha})$ which specify a posterior "$q$" DP distribution, and then applies the evaluation denoising attention function (Equation 4) to the resulting base distribution $G_0^q$. Excluding the prior component (discussed in Section 3.2), we define the projections:

$$\boldsymbol{\mu} = \mu(\mathbf{Z}) = \mathbf{Z}\mathbf{W}^{\mu} + \boldsymbol{b}^{\mu} \tag{6}$$

$$\boldsymbol{\sigma}^2 = \sigma^2(\mathbf{Z}) = \exp(\mathbf{Z}\mathbf{W}^{\sigma} + \boldsymbol{b}^{\sigma}) \tag{7}$$

$$\boldsymbol{\alpha} = \alpha(\mathbf{Z}) = \exp(\mathbf{Z}^2\boldsymbol{w}_1^{\alpha} + \mathbf{Z}\boldsymbol{w}_2^{\alpha} + b^{\alpha}) \tag{8}$$

where $\mathbf{Z}^2$ is the component-wise square. We choose these forms of projections because we want to set its parameters, $\mathbf{W}^{\mu}, \mathbf{W}^{\sigma} \in \mathbb{R}^{d \times d}$, $\boldsymbol{b}^{\mu}, \boldsymbol{b}^{\sigma} \in \mathbb{R}^d$, $\boldsymbol{w}_1^{\alpha}, \boldsymbol{w}_2^{\alpha} \in \mathbb{R}^d$ and $b^{\alpha} \in \mathbb{R}$, so that the evaluation denoising attention function is effectively equivalent to standard attention over $\mathbf{Z}$. For equivalence between standard attention and denoising attention, we must remove the influence of the prior component by down-weighting it, and skew the query-value interpolation to the value by reducing the uncertainty of the non-prior components. Alternatively, we can see this requirement as making the mixture of Gaussians $G_0^q$ equivalent to the mixture of impulse distributions $F_{\mathbf{Z}}$ in Equation 1. See Figure 1 for a visual depiction. We can down-weight the prior component by making the pseudo-counts $\alpha(\mathbf{Z})$ for the non-prior components be large. We can make the non-prior component Gaussians look like the impulse distributions of $F_{\mathbf{Z}}$ by setting the mean projection $\mu(.)$ to be the identity and the variance projection $\sigma^2(.)$ to be approximately zero (exactly zero is not possible with the exponential activation function). Then to make the mixture weights the same as for $F_{\mathbf{Z}}$ in Equation 1, the pseudo-count projection $\alpha(.)$ must be proportional to the exponent of the scaled $L_2^2$ norm of $\mathbf{Z}$, which is why it has a log-quadratic form in Equation 8.

**Initialisation hyperparameters** We define the parameters of these projections to achieve these requirements for equivalence but still allow some uncertainty. Based on preliminary experiments, we propose a small number of initialisation hyperparameters which controls the transition between equivalence and a smooth regularisation of the embeddings (illustrated in Figure 1). The hyperparameters $\tau_{\alpha}^i$ and $\tau_{\sigma}^i$ control the level of uncertainty of the mixture distributions by changing the initialisation of the pseudo-counts and variance, respectively. The indicator $i$ is for the different sections of the model, which allows for independent control of the encoder's self-attention ($e$), decoder's cross-attention ($c$) and decoder's causal self-attention ($d$). Empirically, this allows for more flexibility than a single hyperparameter and is practically easier to tune than defining a hyperparameter per layer. In general we define the layer projection weights as follows:

$$\mathbf{W}^{\mu} = \mathbf{I}; \qquad\qquad \boldsymbol{b}^{\mu} = \mathbf{0}$$

$$\mathbf{W}^{\sigma} = \mathbf{0}; \qquad\qquad \boldsymbol{b}^{\sigma} = \log((\sigma^p * \tau_{\sigma}^i)^2) \tag{9}$$

$$\boldsymbol{w}_1^{\alpha} = \frac{1}{2\sqrt{d/h}} \odot \mathbf{1}; \quad \boldsymbol{w}_2^{\alpha} = \mathbf{0}; \quad b^{\alpha} = \epsilon^{\alpha}\tau_{\alpha}^i \tag{10}$$

where $d$ and $h$ denote the model projection size and number of attention heads. As discussed in the next section, the empirical distribution is used to define the prior standard deviation $\sigma^p$, and the constant $\epsilon^{\alpha}$, which denotes the empirical standard deviation of the

scaled $L_2^2$-norm in log space per layer. The hyperparameter $\tau_\alpha^i$ in Equation 10 controls the relative weight of the prior component to the non-prior components in the mixture distribution $G_0^q$. Since $\epsilon^\alpha$ is a standard deviation, it reflects the normal range of values for the non-prior-component pseudo-counts, so we can use $\tau_\alpha^i$ to control the weight given to the prior component with respect to this range. When $\tau_\alpha^i = 0$, the non-prior pseudo-counts are their scaled $L_2^2$-norms and the prior pseudo-count is the expected scaled $L_2^2$-norm. When we increase or decrease the $\tau_\alpha^i$, it adjusts the magnitude of the non-prior $L_2^2$-norms, which relatively decreases or increases the weight on the prior proportionately to the standard deviation of the scaled $L_2^2$-norm. The hyperparameter $\tau_\sigma$ (Equation 9) controls the interpolation between the query and value (Equation 4), which we set proportionately to the variance of the prior distribution. When $\tau_\sigma \approx 0$, there is effectively no interpolation, as with standard attention. When $\tau_\sigma = 1$, the uncertainty is increased to the level of the empirical prior distribution.

## 3.2 Empirical Priors

The prior distribution is a Dirichlet Process, so it views all vectors in all transformer embeddings as impulses generated from its own base distribution. Therefore, we can estimate the base distribution of the prior empirically by observing the distribution of vectors given forward passes of the transformer model. Taking a Bayesian approach, the NV-Transformer's prior should reflect the distribution over vectors which the pretrained transformer knows before seeing the input text. This is the distribution observed during training. We compute statistics from the fine-tuned corpora and use them to define our priors. We estimate the prior as the best fit of an isotropic Gaussian distribution to the empirical distribution over latent vectors computed when embedding this training corpus.

**Data-informed prior** We can estimate the empirical parameters for an isotropic Gaussian $G_0^p \sim \mathcal{N}(\boldsymbol{\mu}^p, \boldsymbol{I}(\sigma^p))$ and the prior's pseudo-count $\alpha_0^p$ by using the latent vectors $\boldsymbol{Z}$ as follows: $\boldsymbol{\mu}^p = \frac{\sum_i^N \boldsymbol{Z}_i}{N}$, $(\sigma^p)^2 = \frac{\sum_i^N (\boldsymbol{Z}_i - \boldsymbol{\mu}^p)^2}{N-1}$ and $\log(\alpha_0^p) = \sum_i^N \left( \frac{\sum_j^d (\boldsymbol{Z}_{ij})^2}{2\sqrt{d/h}} \right) / N$, where $N$ is the total number of tokens in the training corpus, $d$ is the dimension of the embedding and $h$ is the number of attention heads. This allows the prior mean to be the least informative representation in the center of the latent embeddings vector space. The variance is estimated from this mean. The empirical pseudo-count is kept in log space for numerical stability and is the expected scaled $L_2^2$-norm of the latent vectors.

**Analysis of the empirical prior** To further understand the distributions of our latent embeddings, we calculate the distribution of the empirical priors across all layers of the encoder and decoder attention mechanisms (Appendix F.1). Given a model $\theta_x^*$ that has been trained on data $x$, we analyse priors generated from both in-domain $x$ and out-of-domain summarisation distributions. We find that the distribution of embeddings for the groups: encoder self-attention, decoder cross-attention and decoder causal self-attention, behave differently between and similarly within these attention groups. This observation motivates the choice of grouping the hyperparameters $\tau_\alpha^i$ and $\tau_\sigma^i$ where $i$ is an indicator for encoder, cross or decoder attention mechanisms. We also consider the data requirements of the empirical prior by calculating the prior from a subsample of the training data. We found that the empirical prior can be created from as few as 0.1% of the training data ($\approx 200$ examples) and achieve similar performance (Appendix F.2). Hence, the empirical prior is data-efficient as it requires a minimal amount of training data for creation.

## 4 Post-Training Evaluation

To evaluate our Bayesian reinterpretation of transformers, we first show that our proposed identity initialisation results in empirical equivalence to pretrained transformers. We then investigate how this reinterpretation allows us to apply an information-theoretic, post-training regularisation by changing the initialisation of the NVIB layers. Once the empirical

prior is estimated, adapting a pretrained transformer to a new domain simply requires a hyperparameter selection on data from the new domain, which can be done solely with forward-passes of the model. This is an important contribution as the cost of backward-passes, regularisation and parameter updates becomes increasingly expensive as the models scale. Hence, the experimental design for Nonparametric Variational (NV) regularisation is entirely post-training and would not be fair to compare against fine-tuned methods. We evaluate on the validation set of the original dataset the model was trained on. We refer to this as in-domain. We then evaluate out-of-domain (OOD) generalisation, demonstrating that by changing the initialisation of our model we are able to achieve an information-theoretic post-training regularisation.

**Models** We consider models that are already pretrained and fine-tuned, available on Hug-gingFace Wolf et al. (2020). In our experiments, we do not fine-tune or update the original model weights with data, instead only setting hyperparameters in the initialisation of our NVIB layers, which regularise the attention functions post-training. For a comprehensive application of our attention regulariser, we consider sequence-to-sequence transformer models, which accounts for: encoder self-attention, cross-attention and decoder causal-attention. For summarisation we consider BART Lewis et al. (2020) summarisation models that have already been fine-tuned on either CNN/DailyMail data (See et al., 2017), or Xsum data (Narayan et al., 2018). For translation we consider Marian (Junczys-Dowmunt et al., 2018) encoder-decoder translation models already fine-tuned on the OPUS100 (Zhang et al., 2020) high resource language pairs: English-German (En-De) and English-French (En-Fr). For all tasks we use empirical priors created from their respective fine-tuning corpora. We provide model implementation details in Appendix D.2 and E.2.

## 4.1 Identity Initialisation Equivalence

We show empirically that pretrained transformer models can be interpreted as NV models by showing that our identity initialisation (Section 3.1) does not change their predictions. We set the NVIB initialisation parameters $\tau_\alpha^i{=}10$ and $\tau_\sigma^i \approx 0$ ($1e^{-38}$ for `Float32`) for all $i$. These down-weight the influence of the prior and reduce the interpolation between the query and value in the attention mechanism, respectively (see Equation (4)). Appendix Table 4 shows that the NV-Transformer initialisation has the same accuracy as the pretrained transformer. More precisely, the leftmost points in Figure 2 being at 100% on the $x$-axis shows that these two models generate the same outputs.

**Analysis of Increasing Uncertainty** We show that increasing the uncertainty within our regularisation allows access to models with different outputs but equivalent accuracy. We evaluate increasing the uncertainty of our NV models on the validation sets for the same datasets they were trained on. To characterise the range of these regularised NV models, Figure 2 plots them in a line ordered by their linear interpolations between the equivalent-initialisation and the everything-over-regularised corners of the sample space. The NV-Transformers are plotted by comparing their output using Rouge-L (top) or BLEU (below) against the gold summary or translation ($y$-axis) and against the original non-NV model's output ($x$-axis). As we increase the regularisation from the identity initialisation

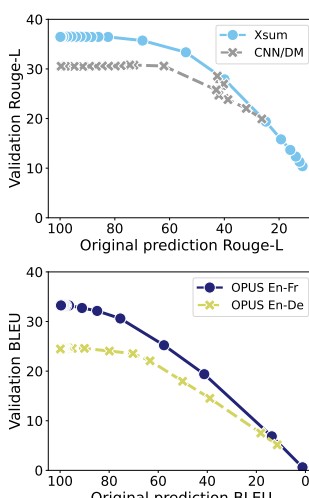

Figure 2: For each NVIB initialisation, the NV-Transformer's output is compared against the non-regularised original model ($x$-axis) and against the gold prediction ($y$-axis). Summarisation (Top), Translation (Below).

we discover a space of models which are not only different from the original baseline model ($x$-axis of $< 100\%$ overlap), but also equally good (same $y$-axis performance). The inclusion of our regularisation gives rise to a smooth transition between these models.

**Attention maps** Examining the attention maps (Appendix F.3) of the summarisation encoder and decoder, we see that the prior component is included in the keys of attention,

Table 1: Post-training regularisation on OOD summarisation. We report test set Rouge-L.

| Model | CNN/DM | Xsum | CC | Out-of-Domain SAMsum | WikiHow |
|---|---|---|---|---|---|
| BART (CNN/DM) | **29.99** | 13.12 | 24.99 | 22.42 | 9.26 |
| BART-16bit (CNN/DM) | 29.97 [-0.02] | 13.13 [+0.01] | 24.99 [ 0.00] | 22.36 [-0.06] | 9.27 [+0.01] |
| BART-8bit (CNN/DM) | 29.55 [-0.44] | 13.13 [+0.01] | 24.67 [-0.32] | 22.13 [-0.29] | 9.36 [+0.10] |
| NV-BART (CNN/DM) | 29.33 [-0.66] | **13.99 [+0.87]** | **25.04 [+0.05]** | **22.60 [+0.18]** | **9.41 [+0.25]** |
| BART (Xsum) | 16.61 | 36.42 | 14.37 | 18.33 | 13.43 |
| BART-16bit (Xsum) | 16.60 [+0.01] | **36.44 [+0.02]** | 14.38 [+0.01] | 18.24 [-0.09] | 13.43 [ 0.00] |
| BART-8bit (Xsum) | 16.56 [-0.05] | 36.25 [-0.17] | 14.33 [-0.04] | 18.05 [-0.28] | 13.53 [+0.10] |
| NV-BART (Xsum) | **19.42 [+2.81]** | 36.25 [-0.17] | **17.61 [+3.24]** | **21.94 [+3.61]** | **15.25 [+1.82]** |

but the identity initialisation down-weights this prior to the point that there is no attention weight to this component. When regularised, attention weight is redirected from tokens such as punctuation, to the prior component.

## 4.2 Out-of-Domain Generalisation

We evaluate whether our post-training information-theoretic regulariser improves out-of-domain performance in the presence of a text domain shift. This is an increasingly important problem as pretrained models become larger and more challenging to fine-tune given a shift in the data domain. To evaluate our regulariser's OOD generalisation we define the following experiment: given a model $\theta_x^*$ that has been trained on a text domain $x$, we evaluate the model on the same task in the presence of a domain-shift to the text domain $y$. For example, given a pretrained news summarisation or translation model $\theta_x^*$ we evaluate the OOD generalisation by shifting the domain from news $x$ to informal dialogues $y$ without further training.

For baselines we consider the original model (32-bit) and quantisation of 16-bit and 8-bit (Dettmers et al., 2022). Quantisation is an alternative form of post-training regularisation which has been shown to improve performance (Dettmers & Zettlemoyer, 2022). We also considered a 4-bit quantisation model and combining NVIB regularisation with 16-bit quantisation (Appendix Table 8, 11). When combined, we found improvements over the original models. We did not consider sparsity techniques as they were not easily available without further training. In the following sections we consider sequence-to-sequence models trained on summarisation and machine translation.

**Summarisation** For our summarisation experiments we evaluate the models on the validation sets of out-of-domain summarisation corpora from either Xsum or CNN/DailyMail, respectively, and from Curation Corpus (CC) (Curation, 2020), SAMsum (Gliwa et al., 2019) and WikiHow (Koupaee & Wang, 2018). We provide further dataset details in Appendix D.1. To evaluate on the test set we select the best model on the validation set (Appendix D.3). Table 1 reports the test set Rouge-L on the OOD text summarisation datasets for the post-training regularisation methods. Considering the original model trained on CNN/-DailyMail, we notice that with NVIB regularisation the OOD performance on the test set improves over the baselines consistently, although minorly. In contrast, the original model trained on Xsum is substantially improved when NVIB regularisation is applied to OOD text summarisation. We speculate the larger improvement in the model trained on Xsum is due to the abstractive nature of the training data, whereas the model trained on CNN/DailyMail is more extractive. This shows that our information-theoretic, post-training regularisation can improve OOD generalisation, both over the original model and over quantisation.

**Analysis of summarisation improvement** We conjecture that the length of the summary is a strong proxy for the information within the document. Since the baseline model has been trained to produce a summary of a certain length, the summaries on OOD data can be improved by adapting the length of the summary to the information content. We

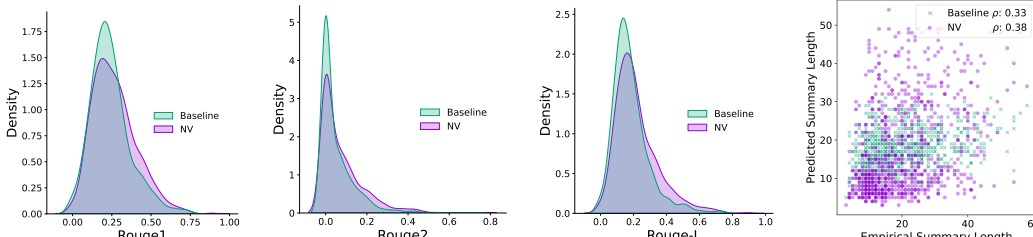

Figure 3: A BART baseline model trained on Xsum compared to its reinterpretation with NVIB regularisation on the SAMsum test dataset. **Left-Right**: Score distributions for Rouge-1, Rouge-2, Rouge-L, and empirical vs predicted summary length with its Spearman's correlation.

see this pattern in the mean sentence lengths in Table 2. The baseline produces summaries slightly shorter than its empirical Xsum training data for all datasets, while the reinterpreted NV-Transformer model matches this length in-domain but produces shorter summaries for shorter-summary datasets and longer summaries for longer-summary datasets. Looking within the distribution of a single OOD dataset, we compare the

Table 2: Mean number of words in validation set summaries for the original BART model trained on Xsum data and a reinterpreted NV-Transformer model.

| Model | CC | CNN/DM | SAMsum | WikiHow | Xsum |
|---|---|---|---|---|---|
| Empirical | 85.2 | 57.9 | 20.3 | 6.5 | 21.1 |
| BART (Xsum) | 19.3 | 20.7 | 17.4 | 16.4 | 18.9 |
| NV-BART (Xsum) | 28.2 | 30.8 | 16.6 | 14.4 | 18.6 |

output summaries of the original BART model and a reinterpreted NV-Transformer model on the OOD SAMsum test dataset. Figure 3 (3 left) shows consistent improvements across the Rouge-1, Rouge-2 and Rouge-L distributions, which measures overlap metrics against the gold summaries. Figure 3 (1 right) plots the output summary lengths against the true empirical summary lengths, and gives the Spearman's correlation coefficient. This suggests that the NV model is better at generalising to the length of the test summary according to the information variation in the input document. In Appendix Figure 5 we provide these plots for all OOD validation datasets. We provide examples of generated summaries of all our summarisation models and in- and out-of-domain datasets in Appendix F.4.

**Translation** We evaluate the translation models on the English to German (En-De) and English to French (En-Fr) validation sets of out-of-domain corpora Bible (Christodoulopoulos & Steedman, 2014), IWSLT (Cettolo et al., 2017) and TedTalks (Cettolo et al., 2012). See Appendix E.1 for further dataset details. We find the best model on the validation set (Appendix 11) and then evaluate this model on the test set. Table 3 reports the test set BLEU on the OOD translation datasets. We notice fairly consistent minor improvements (5 out of 6 cases) across all OOD translation datasets. We speculate that the improvements in translation have a lower magnitude than summarisation, as this task benefits less from a reduction in information. These results further support our hypothesis that our post-training, information-theoretic regulariser improves OOD generalisation of pretrained transformers.

## 5 Discussion

Since regularisation does not guarantee improved performance, it is surprising that our method, which is applied post-training without backpropagation or parameter updates, can improve performance. We believe the variance in performance is attributed to the nature of the different tasks. Summarisation is fundamentally an information compression task, well suited to an information bottleneck, unlike translation. We speculate the larger improvement in summarisation models trained on Xsum is due to the abstractive nature of the dataset. The Xsum summarisation dataset requires more generalisation than the extractive CNN/DailyMail dataset, which relies on selecting a summary from the document.

The improvements from post-training regularisation suggest that our variational Bayesian reinterpretation of pretrained transformers is an accurate model of how information is

Table 3: Post-training regularisation on text translation. We report test set BLEU.

| Model | OPUS100 | Out-of-Domain | | |
| | | Bible | IWSLT | TedTalks |
|---|---|---|---|---|
| Marian (OPUS En-De) | 24.78 | 23.02 | 27.16 | 24.80 |
| Marian-16bit (OPUS En-De) | 24.70 [-0.08] | 23.02 [ 0.00] | 27.18 [+0.02] | 24.79 [-0.01] |
| Marian-8bit (OPUS En-De) | 24.65 [-0.13] | **23.11 [+0.09]** | 27.05 [-0.11] | 24.67 [-0.13] |
| NV-Marian (OPUS En-De) | **24.84 [+0.06]** | 22.95 [-0.07] | **27.30 [+0.14]** | **25.06 [+0.26]** |
| Marian (OPUS En-Fr) | 35.22 | 27.23 | 39.03 | 31.92 |
| Marian-16bit (OPUS En-Fr) | 35.22 [ 0.00] | 27.21 [-0.02] | 39.05 [+0.02] | 31.97 [+0.05] |
| Marian-8bit (OPUS En-Fr) | **35.23 [+0.01]** | 27.19 [-0.04] | 39.03 [0.00] | 31.89 [-0.03] |
| NV-Marian (OPUS En-Fr) | 35.22 [ 0.00] | **27.41 [+0.18]** | **39.28 [+0.25]** | **32.43 [+0.51]** |

being captured in transformer embeddings. When we map an embedding into a probability distribution, the variance specifies what differences between embeddings constitute reliable information, and what differences should be unreliable or unimportant. The denoising attention function then ignores this unreliable information, which should not affect accuracy in-domain, and could reduce overfitting out-of-domain. The above experiments succeed in finding such empirically effective patterns of variance for our proposed DP distributions (Figures 2, 10, 11 and Tables 1, 3). Since this regularisation is done post-training, this argument suggests that the pretrained model has learned what information is unreliable, and that NVIB regularisation exposes this representation of information. In particular, our model suggests that the information in a given embedding dimension value is relative to the distribution over that dimension during training, with the mean value carrying no information and the scale being relative to the variance. They also suggest that the $L_2^2$-norm of a vector reflects the importance of the information in a vector. We anticipate that these insights into how transformer embeddings represent information will help in the understanding and development of future improvements to transformer architectures.

## 6 Conclusion

This work contributes both to the development of novel models which use Nonparametric Variational Information Bottleneck regularisation, and to the understanding of pretrained transformers. Technically, we contributed an extension of NVIB to all forms of multi-head attention, a novel empirical prior, and an effective identity initialisation with controllable hyperparameters. This development of the Nonparametric Variational Transformer architecture allows us to propose a Bayesian reinterpretation of pretrained transformers. Empirically, we analyse and show the usefulness of this smooth, post-training regularisation by improving the performance of existing pretrained transformers in out-of-domain text summarisation and translation, without any fine-tuning. This success suggests that NVIB provides an accurate characterisation of how pretrained transformers represent information in their embeddings.

**Future work** In future work, we plan to evaluate our post-training regularisation on different modalities and tasks, such as: encoder-only classification; and decoder-only language modelling. The success of our regularisation in this post-training scenario suggests that even better generalisation could be achieved by fine-tuning with NVIB regularisation, for which our proposed mapping to a regularised NV-Transformer should be an effective initialisation. With Large Language Models (LLMs), their scale makes it more valuable to regularise them post-training, and makes regularisation more relevant during training. Moreover, understanding how transformer embeddings represent information could improve diversity in LLM generation, since sampling can be done directly in the embeddings of the model. We leave application of NVIB regularisation to LLMs for future work because it is non-trivial to connect denoising attention to relative position encodings (Su et al., 2024) and hardware specific implementations such as flash-attention (Dao et al., 2022).

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

# A  Appendix

## Ethics Statement

This paper presents work whose goal is to advance the field of Machine Learning. There are many potential societal consequences of our work, none of which we feel must be specifically highlighted here.

## Limitations

In this work, we limit our claims and evaluation to only consider the post-training paradigm. In this, we exclude sparsity and further fine-tuned baselines as they require considerable computation due to the parameter updates and backward-passes. Moreover, we limit our choice of models to only consider sequence-to-sequence, encoder-decoder transformers. These models were selected as they comprehensively account for all forms of attention: encoder self-attention, cross-attention and decoder causal-attention. Hence, we exclude decoder-only models, such as many large language models (LLMs). We limit our experimentation to Natural Language Processing (NLP) tasks such as summarisation and translation. Further experimentation in the domain of vision and speech is beyond the scope of this work. We plan to address these limitations in future work.

## B  Denoising Multihead Attention

In this section, we provide the details for *denoising multihead attention* at training and evaluation time. We define the set of transformer latent embedding vectors as $Z \in \mathbb{R}^{n \times d}$ and set of pre-projected queries as $U' \in \mathbb{R}^{m \times d}$. We assume the latent vectors are square such that $W^Q, W^K, W^V \in \mathbb{R}^{d \times d}$ and biases $b^Q, b^K, b^V \in \mathbb{R}^d$ are used to linearly project to the queries, keys and values, respectively. We define the standard attention weights before the softmax as follows:

$$A = \frac{1}{\sqrt{d}} \underbrace{(U'W^Q + b^Q)}_{Q} \underbrace{(ZW^K + b^K)^{\top}}_{K^{\top}} \in \mathbb{R}^{m \times n} \tag{11}$$

Typically, for multihead attention the projected query $Q$ and keys $K$ are split into heads. In this definition, we split the linear projections by a divisible number of heads $h$ such that $W^Q, W^K, W^V \in \mathbb{R}^{h \times d \times \frac{d}{h}}$ and biases $b^Q, b^K, b^V \in \mathbb{R}^{h \times \frac{d}{h}}$, so that $Q \in \mathbb{R}^{h \times m \times \frac{d}{h}}$ and $K \in \mathbb{R}^{h \times n \times \frac{d}{h}}$. We can then specify multihead attention by defining a matrix of attention scores $A \in \mathbb{R}^{h \times m \times n}$, for each head $i$:

$$A_i = \frac{1}{\sqrt{d/h}} ((Q_i (W_i^K)^{\top} Z^{\top} + Q_i (b_i^K)^{\top}))$$

where the bias term $Q_i (b_i^K)^{\top} \in \mathbb{R}^m$ is added across all $n$ keys, and thus is normalised out in the softmax below. The scaling term also considers the heads and is division by $\sqrt{d/h}$.

For denoising attention, each head's query is projected into the space of the original set of vectors $Z$, namely $U_i = Q_i (W_i^K)^{\top}$, and so is still in $\mathbb{R}^{m \times d}$. Thus, each head can be viewed as

doing denoising attention in the same way as single-head attention, with the only difference being that the variance of the theoretical query noise is now $\sqrt{d/h}I$.

## B.1 Training denoising attention

In this work we do not do any training, but for completeness we include the equations for multihead attention at training time. The NVIB layer outputs the isotropic Gaussian parameters $\mu \in \mathbb{R}^{(n+1)\times d}, \sigma \in \mathbb{R}^{(n+1)\times d}$ and Dirichlet parameters $\alpha \in \mathbb{R}^{(n+1)}$, which include the $(n+1)^{\text{th}}$ component for the prior. During training, these are used for sampling such that $\pi \sim \text{Dir}(\alpha)$ and $Z \sim \mathcal{N}(\mu, \sigma)$. Given these sampled weights and vectors, the training-time denoising attention function is the same as the standard attention function with two changes: (1) the keys come from the sampled vectors $Z \in \mathbb{R}^{(n+1)\times d}$, which include a vector sampled from the prior component; and (2) each key has an attention bias $b \in \mathbb{R}^{(n+1)}$ which is determined by its weight $\pi \in \mathbb{R}^{(n+1)}$. Summing over heads $i$, the training-time denoising attention function is:

$$\text{DAttn}(.) = \sum_i \text{Softmax}(A_i + \underbrace{\log(\pi) - \tfrac{1}{2\sqrt{d/h}}\|Z\|^2}_{b})(\underbrace{ZW_i^V + b_i^V}_{V_i}))$$

The biases $b$ are defined by adding the log of the sampled weights $\log(\pi) \in \mathbb{R}^{(n+1)}$ from the NVIB layer and subtracting the scaled $L_2^2$-norms of the sampled vectors $\frac{1}{2\sqrt{d/h}}\|Z\|^2 \in \mathbb{R}^{(n+1)}$. For multihead attention we only need to reuse the same biases $b$ for each head, just like we reuse the same vectors $Z$ for each head.

## B.2 Evaluation denoising attention

During evaluation, as for training, the NVIB layer outputs the isotropic Gaussian parameters $\mu \in \mathbb{R}^{(n+1)\times d}, \sigma \in \mathbb{R}^{(n+1)\times d}$ and Dirichlet parameters $\alpha \in \mathbb{R}^{(n+1)}$. For evaluation, the denoising attention function is similar to the training-time denoising attention function with three changes: (1) the parameters are used directly without sampling; (2) there is a different attention bias $c \in \mathbb{R}^{(n+1)}$ for each of the $(n+1)$ input vectors, including the prior; and (3) the attention values are computed with an interpolation between the query and value. We can write the denoising attention scores $A \in \mathbb{R}^{h \times m \times (n+1)}$, for each head $i$, as follows:

$$A_i = Q_i(W_i^K)^\top \left(\frac{\mu}{\sqrt{d/h + \sigma^2}}\right)^\top + \frac{1}{\sqrt{d/h}}Q_i(b_i^K)^\top$$

where the bias term $Q_i(b_i^K)^\top \in \mathbb{R}^m$ is added across all $n$ keys, and thus is normalised out in the softmax below.

To this attention score matrix $A$, multihead evaluation denoising attention adds the same key biases $c \in \mathbb{R}^{h \times (n+1)}$ across all $m$ queries and $h$ heads. For ease of notation we define $\sigma_r^2 = (\sqrt{d/h} + \sigma^2)$ and note that $\alpha_0 = \sum_{j=1}^d \alpha_j$.

$$c = \log\left(\frac{\alpha}{\alpha_0}\right) - \tfrac{1}{2}\left\|\frac{\mu}{\sigma_r}\right\|^2 - \mathbf{1}_d(\log(\sigma^r))^\top$$

where $\mathbf{1}_d$ is a row vector of $d$ ones.

Now, we define the interpolation of each head's projected query $U_i = Q_i(W_i^K)^\top \in \mathbb{R}^{m \times d}$ with the inference time value vectors $\mu \in \mathbb{R}^{(n+1)\times d}$. The interpolation weights, $\frac{\sigma^2}{(\sqrt{d/h}+\sigma^2)}$ and $\frac{\sqrt{d/h}}{(\sqrt{d/h}+\sigma^2)}$, are specific to values but not to queries, and the attention matrix, $\text{Softmax}(A_i + c)$, maps values to queries, so we can accommodate the interpolation between queries and values by multiplying in the values before the dot product with the attention matrix and multiplying in the queries after this dot product.

$$\text{DAttn}(.) = \sum_i \left(\left(\text{Softmax}(A_i + c)\frac{\sigma^2}{\sigma_r^2}\right) \odot U_i + \text{Softmax}(A_i + c)\left(\frac{\sqrt{d/h}}{\sigma_r^2} \odot \mu\right)\right)W_i^V + b_i^V$$

### B.3 Denoising Self-Attention

We build upon the contribution of Behjati et al. (2023) of including NVIB and denoising attention for encoder self-attention. In that previous work the NVIB regulariser is applied to training single-head self-attention in the stacked layers of a transformer encoder. The queries for denoising self-attention are computed from the original $n$ vectors of the transformer before they are projected to the $(\mu, \sigma, \alpha)$ parameters of NVIB, but the keys and values are computed from the vectors $Z \in \mathbb{R}^{(n+1) \times d}$ of the NVIB layer. This allows every use of the attention function in self-attention to be done with denoising attention. From this previous work we maintain the exponential activation for the pseudo-counts. However, we remove the skip connection between pseudo-counts, because it is not part of the pretrained transformer.

### B.4 Denoising Causal Attention

A causal mask is applied to self-attention in a transformer decoder to simulate a next token prediction when using teacher-forcing (Vaswani et al., 2017). The mask may be applied as before over the attention scores before the softmax. The prior component in the keys is never masked. This prior component acts like a new additional form of start-of-sequence token, without a positional embedding. The bias terms do not depend on which other keys are masked, since the only term which depends on other keys is $\alpha_0$, and this term normalises out in the softmax. Thus, the implementation is identical to denoising self-attention with the inclusion of a diagonal mask.

### B.5 Pseudocode

We provide *pythonic* pseudocode for the following functions: scaled-dot-product attention (Algorithm 1), denoising attention during training (Algorithm 2), and denoising attention during evaluation (Algorithm 3). The purple colour defines the differences between standard scaled-dot-product attention and denoising attention. The @ and * symbols are for matrix multiplication and element wise multiplication, respectively. The ** is used to denote an element-wise power.

**Algorithm 1** Scaled-dot-product Attention

```
class Attention():
    def __init__(self, d, h):
        # Projections to Q, K, V reshaped by heads
        #   [d,d] -> [h,d,d/h]
        self.q = linear(d, d).reshape(h, d, d/h)
        self.k = linear(d, d).reshape(h, d, d/h)
        self.v = linear(d, d).reshape(h, d, d/h)

    def forward(self, u, z):
        # queries      u: [B, M, d]
        # keys / values z: [B, N, d]
        # scale d/h
        scale = keys.shape(2) / self.k.shape(0)

        # Project to Q, K, V
        q = self.q(u)
        k = self.k(z) / sqrt(scale)
        v = self.v(z)

        # Attention scores [h, B, M, N]
        attn = q @ k.transpose()

        # Attention probabilities [h, B, M, N]
        attn = softmax(attn)

        # Value projection [h, B, M, d/h]
        out = attn @ v

        # Reshape [B, M, d]
        out = out.reshape(u.shape())

        return out
```

**Algorithm 2** Denoising Attention (training)

```
class DenoisingAttention():
    def __init__(self, d, h):
        # Projections to Q, K, V reshaped by heads
        #   [d,d] -> [h,d,d/h]
        self.q = linear(d, d).reshape(h, d, d/h)
        self.k = linear(d, d).reshape(h, d, d/h)
        self.v = linear(d, d).reshape(h, d, d/h)

    def forward(self, u, z, pi):
        # queries      u: [B, M, d]
        # keys / values z: [B, N+1, d]
        # scale d/h
        scale = keys.shape(2) / self.k.shape(0)

        # Project to Q, K, V
        q = self.q(u)
        k = self.k(z) / sqrt(scale)
        v = self.v(z)

        # NVIB bias [1, B, 1, N+1]
        b = log(pi) - 1/(2*sqrt(scale))*l2norm(z)**2

        # Attention scores [h, B, M, N+1]
        attn = q @ k.transpose() + b

        # Attention probabilities [h, B, M, N+1]
        attn = softmax(attn)

        # Value projection [h, B, M, d/h]
        out = attn @ v

        # Reshape [B, M, d]
        out = out.reshape(u.shape())

        return out
```

**Algorithm 3** Denoising Attention (evaluation)

```
class DenoisingAttention():
    def __init__(self, d, h):
        # Projections to Q, K, V reshaped by heads
        #   [d,d] -> [h,d,d/h]
        self.q = linear(d, d).reshape(h, d, d/h)
        self.k = linear(d, d).reshape(h, d, d/h)
        self.v = linear(d, d).reshape(h, d, d/h)

    def forward(self, u, mu, sigma2, alpha):
        # queries        u: [B, M, d]
        # keys / values mu: [B, N+1, d]
        scale = mu.shape(2) / self.k.shape(0)

        # Project to Q, K, V
        q = self.q(u)
        k = self.k(mu / (sqrt(scale)+sigma2))
        # v is used in interpolation

        # NVIB bias [1, B, 1, N+1]
        b = log(alpha / sum(alpha))
            - 1/(2*(sqrt(scale)+sigma2))*l2norm(mu)**2
            - sum(log(sqrt(sqrt(scale)+sigma2)))

        # Attention scores [h, B, M, N+1]
        attn = q @ k.transpose() + b

        # Attention probabilities [h, B, M, N+1]
        attn = softmax(attn)

        # Query projection to key-space
        # [h, B, M, d/h] -> [h, B, M, d]
        u_k = self.k(q)

        # Value interpolation [h, B, M, d]
        out = (attn @ (sigma2/(sqrt(d)+sigma2)))*u_k
            + attn @ ((sqrt(d)/(sqrt(d)+sigma2)))*mu

        # Project into self.v space
        # [h, B, M, d] -> [h, B, M, d/h]
        out = self.v(out)

        # Reshape [B, M, d]
        out = out.reshape(u.shape())

        return out
```

## C    Empirical Equivalence

Table 4: NV-Transformer equivalence to pretrained transformers. Validation cross-entropy (CE), Rouge-L scores for in-domain text summarisation (above) and machine translation BLEU (below).

| Data | CE | Rouge-L / BLEU |
|---|---|---|
| BART (CNN/DM) | 2.71 | 30.56 |
| NV-BART (CNN/DM) | 2.71 [ 0.00] | 30.56 [ 0.00] |
| BART (Xsum) | 2.30 | 36.47 |
| NV-BART (Xsum) | 2.30 [ 0.00] | 36.47 [ 0.00] |
| Marian (OPUS En-De) | 1.87 | 24.43 |
| NV-Marian (OPUS En-De) | 1.87 [ 0.00] | 24.43 [ 0.00] |
| Marian (OPUS En-Fr) | 1.45 | 33.25 |
| NV-Marian (OPUS En-Fr) | 1.45 [ 0.00] | 33.25 [ 0.00] |

Table 5: Dataset statistics.

| Dataset | Examples | | | Mean words | |
| | Train | Val | Test | Document | Summary |
|---|---|---|---|---|---|
| CNN/DailyMail | 287K | 13.4K | 11.5K | 685 | 52 |
| Xsum | 204K | 11.3K | 11.3K | 431 | 23 |
| Curation Corpus | 15K | 7.5K | 7.5K | 504 | 83 |
| SAMsum | 14.7K | 0.8K | 0.8K | 94 | 20 |
| Wikihow | 198K | 6K | 6K | 580 | 62 |

## D   Summarisation Experimental Setup

### D.1   Data

In this work we use commonly available summarisation datasets from HuggingFace[2]. We use the following datasets: **CNN/DailyMail** (**CNN/DM**) (See et al., 2017) which is one of the most widely used summarization corpora. It is based on news articles from the CNN and DailyMail websites. The summary sentences are a concatenation of human-generated "highlights" and bullet points. **Xsum** (Narayan et al., 2018) is abbreviated for the "extreme summarization" dataset and is created from BBC news articles. The summaries are taken to be the first sentence of the article and the source document is the rest of the article. **Curation Corpus** (**CC**) (Curation, 2020) is a dataset of professionally written summaries of news articles. This is the only freely available news summarization dataset with references that were written for the purpose of summarizing the article. For this dataset we use the version available on HuggingFace and split it manually into train, validation and test by a 50%/25%/25% split. **SAMsum** (Gliwa et al., 2019) is an abstractive dialogue summarization dataset which is constructed to resemble the chats of a mobile messenger app. Each dialogue is written by a single linguist which can be formal or informal, and potentially contains slang, emoticons or typos. **WikiHow** was constructed from a knowledge base of how-to articles, explaining how to solve a task. We use the text as the document and the headline as the summary. Table 5 provides dataset statistics.

### D.2   Models

In this work we consider the BART encoder-decoder model (Lewis et al., 2020), which is a transformer-based sequence-to-sequence model and is pretrained as a denoising autoencoder. This model has shown efficacy in a wide range of tasks including summarisation. In our work we consider BART (large) summarisation models that are already pretrained and fine-tuned on CNN/DailyMail[3] and Xsum[4], available on HuggingFace. We do not fine-tune or update the original model weights; we only initialise our NVIB layer as a form of post-training regularisation.

The BART (large) model uses a 12 layer transformer encoder and decoder with 16 attention-heads. The size for the word embedding vectors and model projections are 1024, and the size of the feed forward dimensions are 4096, which leads to models of approximately 406 million parameters. The inclusion of the NVIB projection layers, per attention mechanism, results in an increase of approximately 11% parameters. These projections are not trained and only initialised, which results in 459 million parameters in total. It has an input context length of 1024 tokens. Where the tokens are created from a Byte-Pair-Encoding (BPE) tokenizer. Table 6 provides specific autoregressive generation details for each of the models.

---

[2]https://huggingface.co/datasets

[3]https://huggingface.co/facebook/bart-large-cnn

[4]https://huggingface.co/facebook/bart-large-xsum

Table 6: BART models specific autoregressive generation details.

|  | BART (CNN/DM) | BART (Xsum) |
|---|---|---|
| Number of beams | 4 | 6 |
| Length penalty | 2 | - |
| Max length | 142 | 62 |
| Min length | 56 | 11 |

### D.3 Validation Hyperparameters

To get final evaluation scores, we first decrease the search space of NVIB hyperparmeters by finding the points at which each hyperparameter individually has full equivalence and has degradation in performance. We record this space of parameters in Table 7.

Table 7: BART model's NVIB hyperparmeter selection space for random search.

|  | $\tau_\alpha^e$ | $\tau_\alpha^c$ | $\tau_\alpha^d$ | $\tau_\sigma^e$ | $\tau_\sigma^c$ | $\tau_\sigma^d$ |
|---|---|---|---|---|---|---|
| min | -10 | -15 | 1 | $1e^{-38}$ | $1e^{-38}$ | $1e^{-38}$ |
| max | 0 | 0 | 5 | 0.5 | 0.5 | 0.5 |

We notice that $\tau_\alpha^e$ and $\tau_\alpha^c$ can be decreased by several standard deviations before the noise affects the performance. We also notice that the $\tau_\alpha^d$ range shows that the decoder is more sensitive to this parameter. The interpolation parameters $\tau_\sigma$ have about the same sensitivity across the encoder, cross attention and decoder. The best hyperparamters for each model and each validation dataset are visualised in Figure 4.

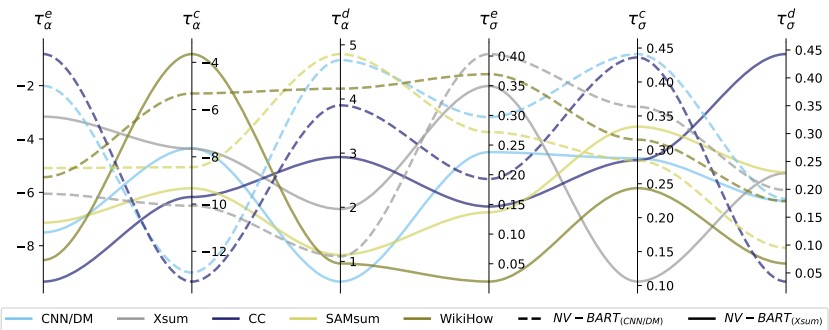

Figure 4: Parallel coordinate plots of best hyperparemeters across models and validation datasets.

After finding the hyperparameter range, we perform a random search of 50 trials for each dataset to find the best regularised models. Table 8 reports the validation results on the out-of-domain text summarisation task for BART models with post-training regularisation methods, including quantisation and NVIB regularisation. For quantisation we consider 16-bit, 8-bit and 4-bit baselines. We also include a combination of NVIB regularisation with quantisation, but the implementation currently only supports 16-bit.

In Figure 5, we compare each predicted validation summary to its corresponding gold empirical summary and report the distribution of Rouge overlap scores and correlations in length. Comparing these distributions for the baseline and for the NV-Transformer tells us in what way we are getting improvements. Comparing the lengths tells us how well the information content of the summary is being modelled. The summary is a compressed and high information version of the original document. The summary length is a measure of information in the document such that a longer summary means more information in a summary, given the compression ratio of the dataset. We compare the predicted and

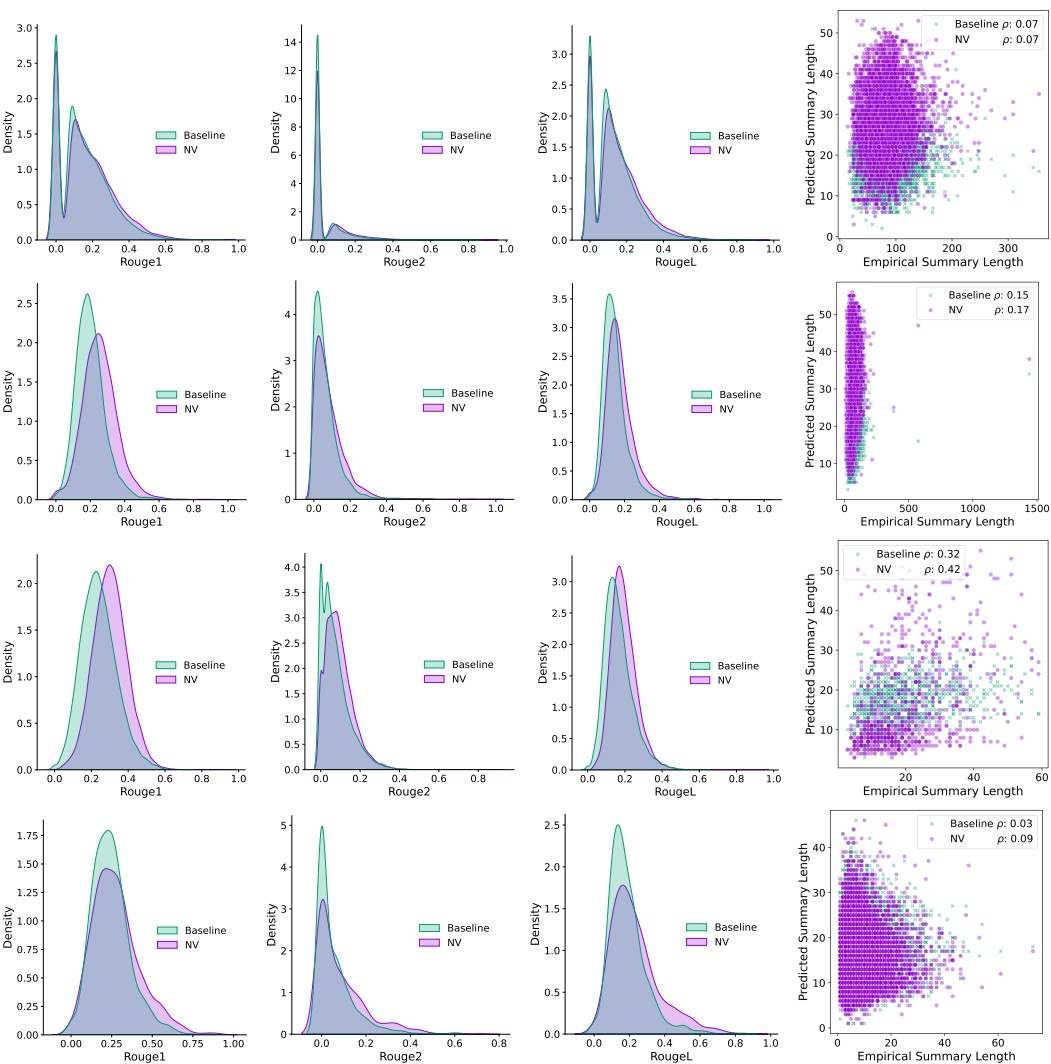

Figure 5: A baseline BART (Xsum) is compared against the same reinterpreted model with NVIB regularisation on **Top-Bottom** Curation Corpus, CNN/DailyMail, SAMsum and WikiHow validation datasets. **Left-Right**: Rouge-1, Rouge-2, and Rouge-L score distributions; Empirical vs predicted summary length with Spearman's correlation.

Table 8: Post-training regularisation on OOD text summarisation. We report validation set Rouge-L.

| Model | CNN/DM | Xsum | CC | Out-of-Domain SAMsum | WikiHow |
|---|---|---|---|---|---|
| BART (CNN/DM) | 30.56 | 13.12 | 25.24 | 23.11 | 9.19 |
| BART-16bit (CNN/DM) | 30.55 [-0.01] | 13.13 [+0.01] | 25.25 [+0.01] | 23.11 [ 0.00] | 9.20 [+0.01] |
| BART-8bit (CNN/DM) | 30.47 [-0.09] | 13.05 [-0.07] | 25.01 [-0.23] | 23.22 [+0.11] | 9.17 [-0.02] |
| BART-4bit (CNN/DM) | 30.33 [-0.23] | 13.14 [+0.02] | 24.78 [-0.46] | 22.51 [-0.60] | 9.17 [-0.02] |
| NV-BART-16bit (CNN/DM) | 29.70 [-0.86] | **14.05 [+0.93]** | 25.38 [+0.14] | 23.38 [+0.27] | **9.40 [+0.21]** |
| NV-BART (CNN/DM) | **30.80 [+0.24]** | 14.00 [+0.88] | **25.46 [+0.22]** | **23.57 [+0.46]** | 9.37 [+0.18] |
| BART (Xsum) | 16.57 | 36.47 | 14.41 | 18.68 | 13.35 |
| BART-16bit (Xsum) | 16.57 [ 0.00] | **36.48 [+0.01]** | 14.42 [+0.01] | 18.67 [-0.01] | 13.35 [ 0.00] |
| BART-8bit (Xsum) | 16.53 [-0.04] | 35.78 [-0.69] | 14.42 [+0.01] | 17.84 [-0.84] | 13.14 [-0.21] |
| BART-4bit (Xsum) | 16.39 [- 0.18] | 35.05 [-1.42] | 14.46 [+0.04] | 16.45 [-2.23] | 13.05 [-0.20] |
| NV-BART-16bit (Xsum) | 18.96 [+2.39] | 36.22 [-0.25] | 17.43 [+3.02] | **23.31 [+4.63]** | **15.06 [+1.71]** |
| NV-BART (Xsum) | **19.43 [+2.86]** | 36.45 [-0.02] | **17.70 [+3.39]** | 23.29 [+4.61] | 14.96 [+1.61] |

true summary length and compute their Spearman's correlation coefficent. If the summary lengths are more correlated than the baseline, it shows the regularisation is adjusting the length according to the information in the article.

Figure 5 shows plots for the model trained on Xsum. The first two rows show that NVIB regularisation increases the performance on the Curation Corpus and CNN/DailyMail datasets and is producing longer, more accurate summaries, but the adaptation to the information in the document is similar to the baseline. However, NVIB regularisation is resulting in longer summaries than the BART (Xsum) model, which better reflects in absolute terms the much longer summaries in the Curation Corpus dataset (See Table 2). The last two rows show that NVIB regularisation increases the performance on the SAMsum and WikiHow datasets and produces shorter, more accurate summaries that are adaptive to the information.

# E    Translation Experimental Setup

## E.1    Data

In this work we use commonly available translation datasets from HuggingFace[5]. We consider high-resource language pairs English-German (En-De), English to French (En-Fr) for our translation experiment with the in-domain training data being the **OPUS100** dataset (Zhang et al., 2020). This is a publicly available parallel corpus with a wide domain. For out-of-domain translation corpora we consider the **Bible** (Christodoulopoulos & Steedman, 2014), **IWSLT** translation dataset from 2017 (Cettolo et al., 2017) and **TedTalks** (Cettolo et al., 2012) datasets. For consistency we randomly split the OOD data into train, validation and test. For the small TedTalks dataset we use an even split. Since the IWSLT dataset was already split, we increased the validation data from 1K examples to 10K from the training data. Finally, from the Bible data we split the training data into validation (10K) and test (10K) as seen in Table 9 below.

## E.2    Models

In this work we consider the Marian encoder-decoder model (Junczys-Dowmunt et al., 2018), which is a transformer-based sequence-to-sequence model and is fine-tuned on the

---

[5]https://huggingface.co/datasets

Table 9: Dataset statistics.

| Dataset | Train | | Validation | | Test | |
|---|---|---|---|---|---|---|
| | En-De | En-Fr | En-De | En-Fr | En-De | En-Fr |
| OPUS100 | 1000K | 1000K | 2K | 2K | 1K | 1K |
| Bible | 42K | 42K | 10K | 10K | 10K | 10K |
| IWSLT | 197K | 224K | 10K | 10K | 8K | 8K |
| TedTalks | 1K | 1K | 1K | 1K | 1K | 1K |

OPUS dataset in specific monolingual language pairs En-De[6] and En-Fr[7], available on HuggingFace. We do not fine-tune or update the original model weights; we only initialise our NVIB layer as a form of post-training regularisation.

The Marian models use a 6 layer transformer encoder and decoder with 8 attention-heads. The size for the word embedding vectors and model projections are 512, and the size of the feed forward dimensions are 2048, which leads to models of approximately 74 million parameters. The inclusion of the NVIB projection layers, per attention mechanism, results in an increase of approximately 9% parameters. These projections are not trained and only initialised, which results in 81 million parameters in total. It has an input context length of 512 tokens. Where the tokens are created from a SentencePiece tokenizer. For autoregressive generation these models use beam search with 4 beams, and a max length of 512.

### E.3 Validation Hyperparameters

To get final evaluation scores we first decrease the search space of NVIB hyperparmeters. We find the points at which each hyperparameter individually has full equivalence and has degradation in performance for each model and each dataset.

Table 10: Marian model's NVIB hyperparmeter selection space for random search.

| | $\tau_\alpha^e$ | $\tau_\alpha^c$ | $\tau_\alpha^d$ | $\tau_\sigma^e$ | $\tau_\sigma^c$ | $\tau_\sigma^d$ |
|---|---|---|---|---|---|---|
| min | -2 | -7 | 0 | $1e^{-38}$ | $1e^{-38}$ | $1e^{-38}$ |
| max | 5 | 10 | 5 | 0.05 | 0.8 | 0.3 |

We notice that $\tau_\alpha^e$ and $\tau_\alpha^c$ can be decreased by several standard deviations before the noise affects the performance. We also notice that the $\tau_\alpha^d$ range shows that the decoder is more sensitive to this parameter. The interpolation parameters $\tau_\sigma^i$ vary a lot with their sensitivity for translation, especially for the encoder. After finding a suitable range for the hyperparameters, we perform a random search of 100 trials for each dataset to find the best regularised models. The best hyperparamters for each model and each validation dataset are visualised in Figure 6.

Table 11 reports the validation results on the out-of-domain translation with original baselines and NVIB regularisation.

---

[6]https://huggingface.co/Helsinki-NLP/opus-mt-en-de
[7]https://huggingface.co/Helsinki-NLP/opus-mt-en-fr

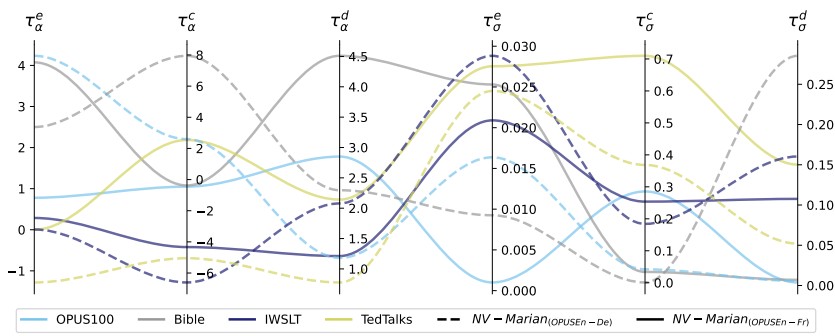

Figure 6: Parallel coordinate plots of best hyperparemeters across models and validation datasets.

Table 11: Post-training regularisation on text translation. We report validation set BLEU.

| Model | OPUS100 | Bible | IWSLT | TedTalks |
|---|---|---|---|---|
| | | Out-of-Domain | | |
| Marian (OPUS En-De) | 24.43 | 23.61 | 24.34 | 26.90 |
| Marian-16bit (OPUS En-De) | 24.44 [+0.01] | 23.64 [+0.03] | 24.36 [+0.02] | 26.85 [-0.05] |
| Marian-8bit (OPUS En-De) | 24.33 [-0.10] | **23.69 [+0.08]** | 24.37 [+0.03] | 27.01 [+0.11] |
| Marian-4bit (OPUS En-De) | 23.89 [-0.54] | 22.57 [-1.04] | 23.79 [-0.55] | 26.77 [-0.13] |
| NV-Marian-16bit (OPUS En-De) | 24.11 [-0.32] | 23.56 [-0.05] | 24.50 [+0.16] | 27.26 [+0.36] |
| NV-Marian (OPUS En-De) | **24.67 [+0.24]** | 23.64 [+0.03] | **24.52 [+0.18]** | **27.28 [+0.38]** |
| Marian (OPUS En-Fr) | 33.25 | 27.15 | 34.04 | 31.38 |
| Marian-16bit (OPUS En-Fr) | 33.23 [-0.02] | 27.16 [+0.01] | 34.04 [ 0.00] | 31.35 [-0.03] |
| Marian-8bit (OPUS En-Fr) | 33.26 [+0.01] | 27.16 [+0.01] | 34.00 [-0.04] | 31.38 [ 0.00] |
| Marian-4bit (OPUS En-Fr) | 32.89 [-0.36] | 25.09 [-2.06] | 33.58 [-0.46] | 30.84 [-0.54] |
| NV-Marian-16bit (OPUS En-Fr) | 33.27 [+0.02] | 27.20 [+0.05] | **34.21 [+0.17]** | 31.66 [+0.28] |
| NV-Marian (OPUS En-Fr) | **33.30 [+0.05]** | **27.39 [+0.24]** | 34.19 [+0.15] | **31.89 [+0.51]** |

# F  Additional Analyses

## F.1  Empirical Prior Analysis

We conduct an analysis of how the distribution over vectors changes through the layers and with different datasets, by calculating the empirical prior parameters for each distribution. We consider the BART model trained on CNN/DailyMail and Xsum and calculate the empirical priors $(\mu^p, \sigma^p, \alpha_0^p)$ given different in- and out-of-domain datasets. The empirical prior used for NVIB regularisation is the one computed in-domain. We also include a standard normal Gaussian prior in the plots for reference. We plot the average embedding value across all layers, where the last layer of the encoder (Encoder layer 13) is the embedding for cross-attention.

Figure 7 displays the average empirical embedding across all encoder layers for a model fine-tuned on CNN/DailyMail and Xsum, respectively. We notice that the mean is approximately zero and very similar across datasets and across models. The variance is close to 0.1 with the cross-attention (Encoder layer 13) being lower at 0.02. The expectation of the logged pseudo-count is higher and approximately around 7 for encoder layers and lower around 2 for cross attention (Encoder layer 13) with consistent variation across datasets. Its clear the distribution of embeddings between fine-tuned BART encoders is similar.

Figure 8 shows the average empirical embedding across all decoder layers for a model fine-tuned on CNN/DailyMail and Xsum, respectively. The decoder means have larger values in comparison to the encoder and similar low variance. Considering the logged

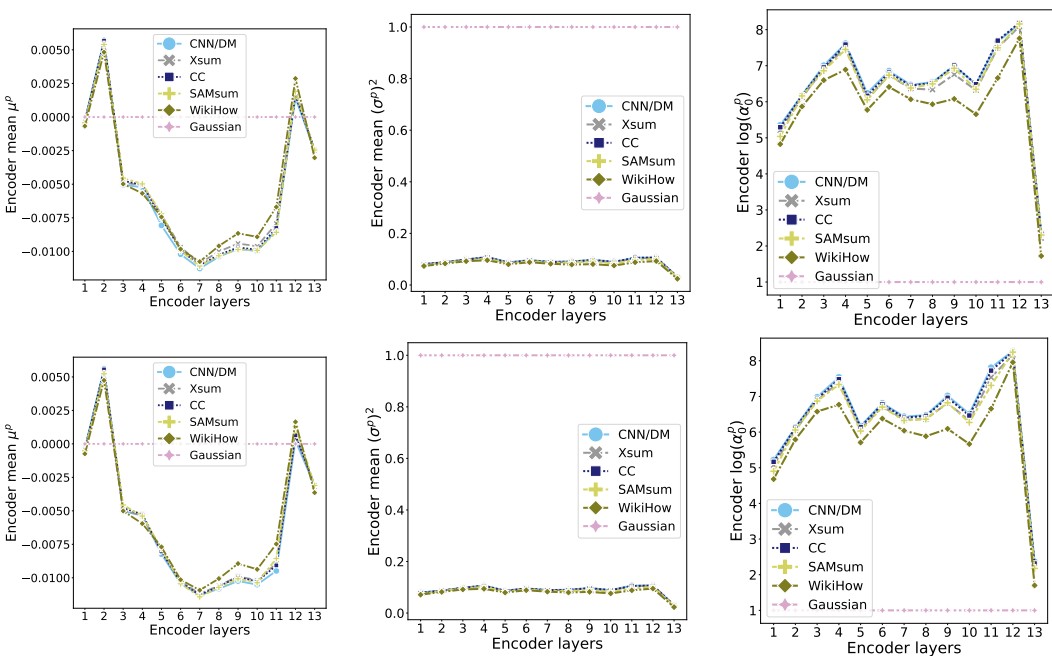

Figure 7: **Top**: BART fine-tuned on CNN/DailyMail. **Bottom**: BART fine-tuned on Xsum. Averaged empirical embeddings per layer **Left:** encoder mean component $\mu^p$. **Middle:** encoder variance $(\sigma^p)^2$. **Right:** encoder logged pseudo-count $\log(\alpha_0^p)$.

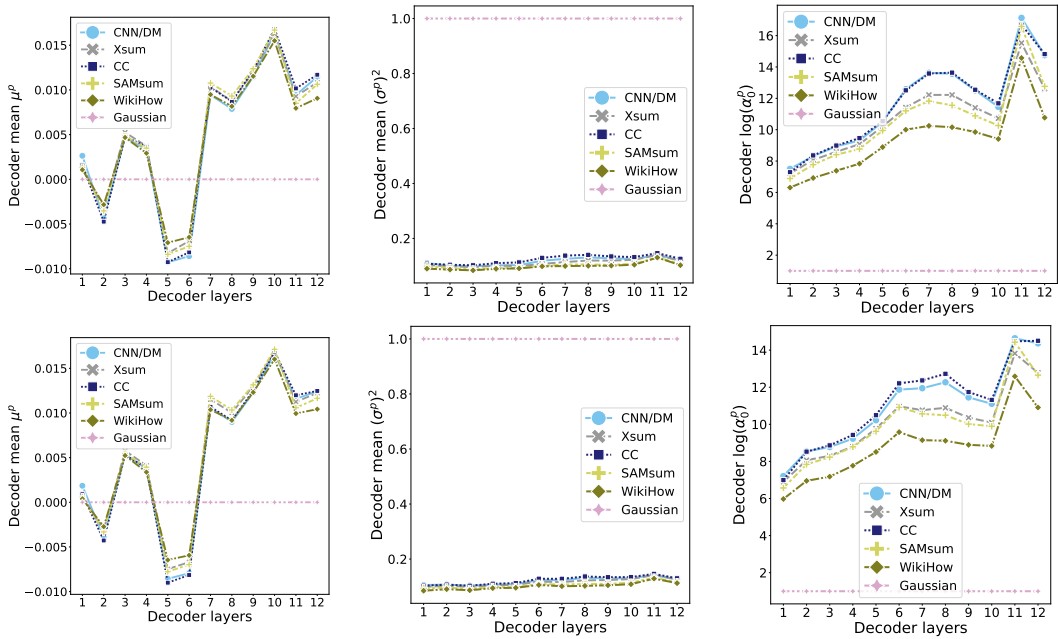

Figure 8: **Top**: BART fine-tuned on CNN/DailyMail. **Bottom**: BART fine-tuned on Xsum. Averaged empirical embeddings per layer of **Left:** decoder mean component $\mu^p$, **Middle:** decoder variance $(\sigma^p)^2$, **Right:** decoder logged pseudo-count $\log(\alpha_0^p)$. "Gaussian" is a unit Gaussian, for reference.

pseudo-count we notice that they get exceedingly large through the layers which are near double the magnitude to the encoder in log-space.

## F.2 Empirical prior data requirements

In this section we restrict the amount of data used to create the empirical prior and see if it is still able to maintain the performance increase across the datasets. We plot the Rouge-L performance as a function of the amount of data used to create the prior. We consider the best selected hyperparameters from the validation set (Section D.3) and show the relative performance for all models and all datasets. Figure 9 shows that the empirical prior is data efficient as it requires few examples (0.1% of training data which is approximately 200) to achieve good stable performance.

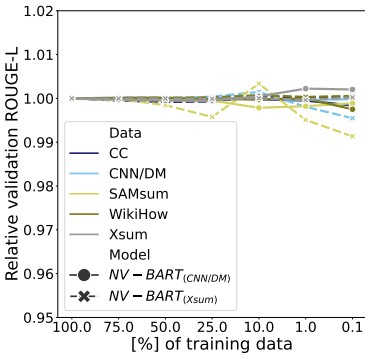

Figure 9: The relative validation performance of Rouge-L ($y$-axis) is compared for different amounts of training data used to create the empirical prior ($x$-axis).

## F.3 Attention plots

In this section we consider the attention plots of our reinterpreted model in the presence of different levels of regularisation. We selected a random validation example from the Curation Corpus dataset and shorten the article document for visualisation purposes. We manually selected the layers with the most regularisation towards the prior for best visualisation. The attention plots are averaged over all heads and the attention scores are displayed with light yellow for 1 and dark purple for 0.

We show an examples of encoder self attention, decoder cross attention and decoder causal attention for both reinterpreted NV-Transformers. We consider three cases: firstly the identity initialisation $\tau_\sigma^i \approx 0$ and $\tau_\alpha^i = 10$; secondly with regularisation where the $\tau_\sigma^i$ and $\tau_\alpha^i$ are set to the best validation hyperparameters (Appendix D.3); and finally an over-regularised example to show what happens in the presence of collapse to the prior component $[P]$, where the initialisation parameters are $\tau_\sigma^i \approx 0$ and $\tau_\alpha^i = -30$. The prior collapse case can be interpreted as the non-prior pseudo-counts being 30 standard deviations smaller than the prior.

In Figures 10, 11 and 12 we see that when the model uses the identity initialisation the attention maps completely ignore the prior component representation $[P]$ and has equivalence in attention scores across all attention functions. In contrast, when $\tau_\alpha^i = -30$ the model has over emphasized the importance of the prior component, such that the attention patterns collapse to only considering the prior. When the model is regularised according to the best validation performance, we notice the attention patterns generally shift attention away from the vertical bars at special characters like punctuation and towards the prior. This shows that the models that are getting improvements have attention distributions which are regularised towards the prior.

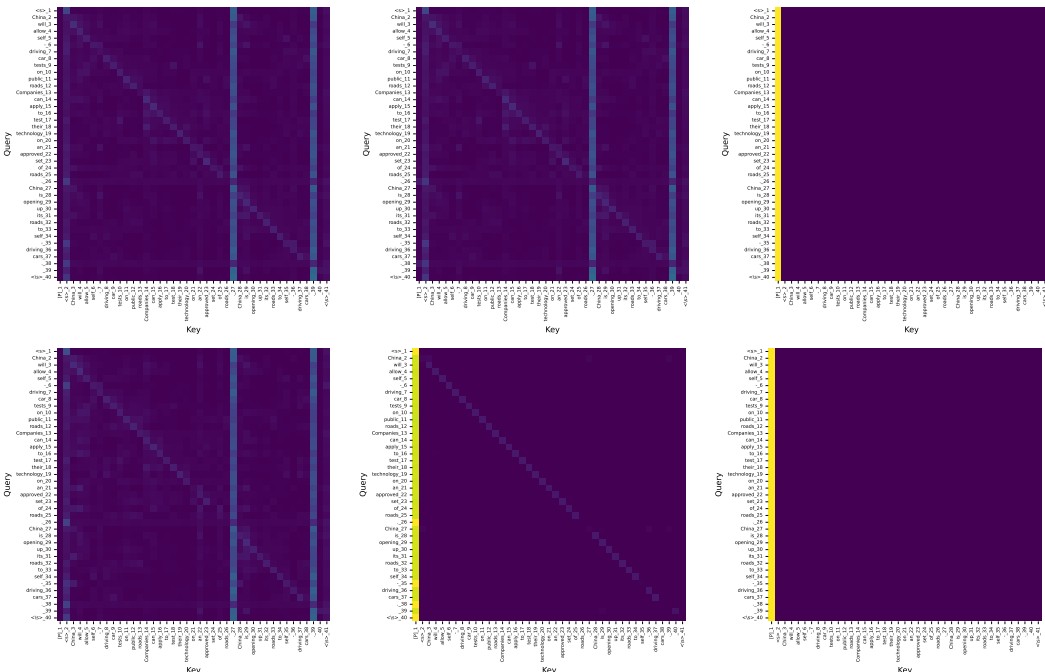

Figure 10: Encoder at layer 10 averaged over all heads self-attention maps. **Top:** CNN/Dailymail. **Bottom:** Xsum. **Left:** Equivalence initialisation. **Middle:** Best post-training NVIB regularisation from validation set **Right:** Prior collapse.

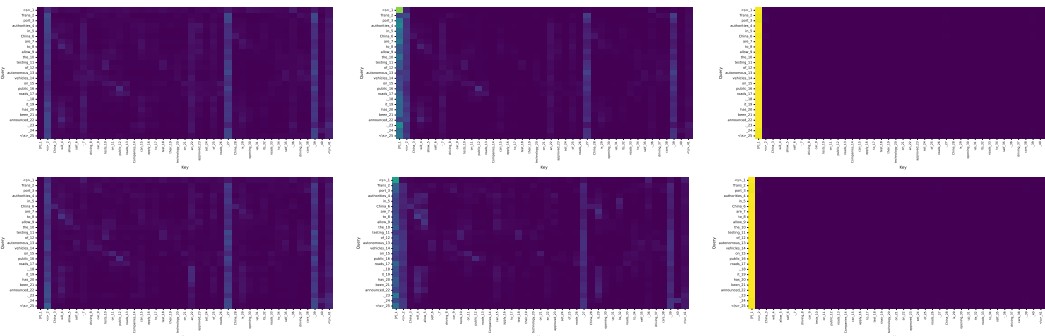

Figure 11: Decoder cross-attention maps at layer 3 averaged over all heads. **Top:** CNN/Dailymail. **Bottom:** Xsum. **Left:** Equivalence initialisation. **Middle:** Best post-training NVIB regularisation from validation set **Right:** Prior collapse.

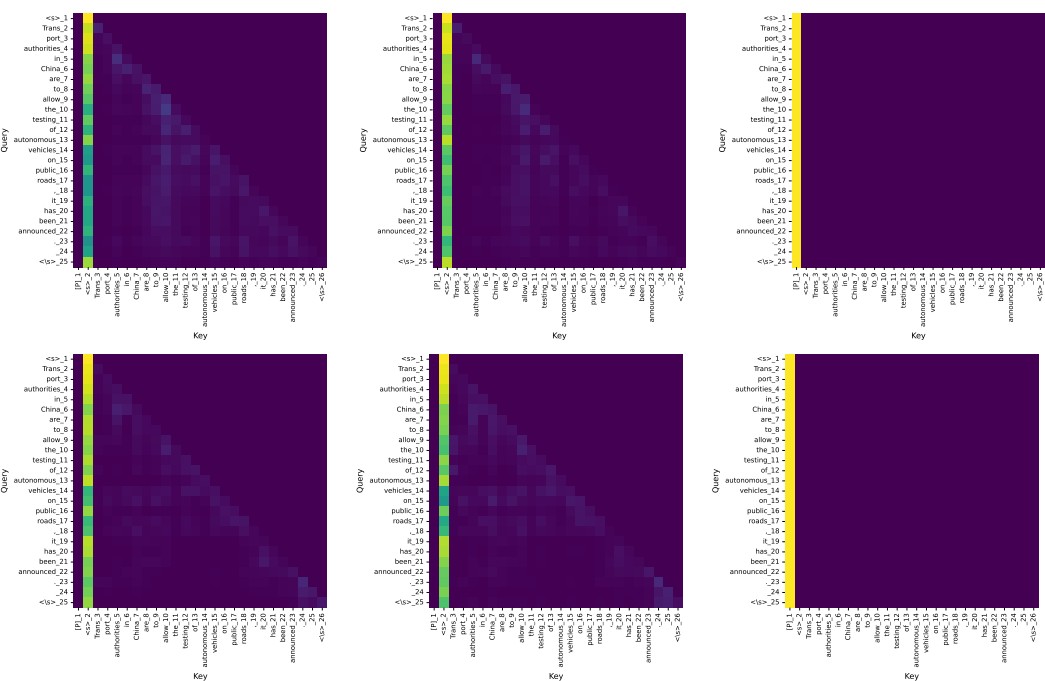

Figure 12: Decoder causal self-attention maps averaged over all heads. **Top:** CNN/-Dailymail layer 6. **Bottom:** Xsum layer 6. **Left:** Equivalence initialisation. **Middle:** Best post-training NVIB regularisation from validation set **Right:** Prior collapse.

### F.4 Generated outputs

In this section we provide example output summaries from our original baseline and reinterpreted NV models for the first example of each validation dataset. Each model only has a context of 1024 tokens in the BART model. We see that the models with post-training regularisation produce similar but different and sometimes improved predictions.

---

**SAMsum**

**Article** A: Hi Tom, are you busy tomorrow's afternoon? B: I'm pretty sure I am. What's up? A: Can you go with me to the animal shelter?. B: What do you want to do? A: I want to get a puppy for my son. B: That will make him so happy. A: Yeah, we've discussed it many times. I think he's ready now. B: That's good. Raising a dog is a tough issue. Like having a baby ;-) A: I'll get him one of those little dogs. B: One that won't grow up too big;-) A: And eat too much;-)) B: Do you know which one he would like? A: Oh, yes, I took him there last Monday. He showed me one that he really liked. B: I bet you had to drag him away. A: He wanted to take it home right away ;-). B: I wonder what he'll name it. A: He said he'd name it after his dead hamster – Lemmy - he's a great Motorhead fan :-))

**Summary** A will go to the animal shelter tomorrow to get a puppy for her son. They already visited the shelter last Monday and the son chose the puppy.

**BART** (CNN/DM) A mother wants to get a puppy for her son. She asks her son to accompany her to the animal shelter to buy him a dog. The boy takes the dog home and names it Lemmy after his dead hamster – Lemmy - a Motorhead fan. The dog is taken back to the shelter.

**NV-BART** (CNN/DM) A mother wants to get a puppy for her son. She asks her son to take her to the animal shelter. He wants a little dog. She says he wanted to take it home right away. He said he'd name it after his dead hamster – Lemmy.

**BART** (Xsum) A conversation between a mother and her son has been caught on camera, in which the mother says she wants to get a puppy for her son.

**NV-BART** (Xsum) I've been chatting to my friend Tom and he's got a puppy for his son and we're planning to get him a dog.

---

**XSum**

**Article** The ex-Reading defender denied fraudulent trading charges relating to the Sodje Sports Foundation - a charity to raise money for Nigerian sport. Mr Sodje, 37, is jointly charged with elder brothers Efe, 44, Bright, 50 and Stephen, 42. Appearing at the Old Bailey earlier, all four denied the offence. The charge relates to offences which allegedly took place between 2008 and 2014. Sam, from Kent, Efe and Bright, of Greater Manchester, and Stephen, from Bexley, are due to stand trial in July. They were all released on bail.

**Summary** Former Premier League footballer Sam Sodje has appeared in court alongside three brothers accused of charity fraud.

**BART** (CNN/DM) Sam Sodje, 37, is jointly charged with elder brothers Efe, 44, Bright, 50 and Stephen, 42. The charge relates to offences which allegedly took place between 2008 and 2014. Sam, from Kent, Efe and Bright, of Greater Manchester, and Stephen,. from Bexley, are due to stand trial in July.

**NV-BART** (CNN/DM) the Sodje Sports Foundation - a charity to raise money for Nigerian sport. Mr Sodje, 37, is jointly charged with elder brothers Efe, 44, Bright, 50, and Stephen, 42. The four deny fraudulent trading charges relating to the alleged offences between 2008 and 2014.

**BART** (Xsum) Former Premier League footballer Sam Sodje has appeared in court charged with fraud.

**NV-BART** (Xsum) Former Premier League footballer Sam Sodje has appeared in court charged with fraud.

---

**Curation Corpus**

---

**Article** China will allow self-driving car tests on public roads Companies can apply to test their technology on an approved set of roads.China is opening up its roads to self-driving cars. The Beijing Municipal Transport Commission released a statement today saying that on certain roads and under certain conditions, companies registered in China will be able to test their autonomous vehicles. Last year, Chinese authorities banned self-driving vehicles from the country's highways until new regulations could be created and approved. But those have been slow to arrive, which is why Chinese company Baidu and its CEO Robin Li came under fire this summer for conducting an apparently illegal demonstration of its driverless technology. The new regulations will allow companies to apply for temporary permission to test their autonomous vehicles on a yet-to-be-determined group of approved roads. The cars will have to have traffic accident liability insurance and a human behind the wheel to take over if anything goes wrong. With this development, China now joins a number of other countries allowing self-driving technology developers to test their products in real-life scenarios.

**Summary** iThe Beijing Municipal Transport Commission will license companies based in China to conduct tests of self-driving cars on some roads under certain conditions, set out in new regulations. These include a requirement for traffic accident insurance and for a human back-up driver to be behind the wheel to take over if necessary.

**BART** (CNN/DM) China will allow self-driving car tests on public roads. Companies can apply to test their technology on an approved set of roads. The cars will have to have traffic accident liability insurance and a human behind the wheel to take over if anything goes wrong. China now joins a number of other countries allowingSelf-driving technology developers totest their products in real-life scenarios.

**NV-BART** (CNN/DM) China will allow self-driving car tests on public roads. Companies can apply to test their technology on an approved set of roads. The cars will have to have traffic accident liability insurance and a human behind the wheel to take over if anything goes wrong. China joins a number of other countries allowing technology developers to test products in real-life scenarios.

**BART** (Xsum) China is opening up its roads to self-driving cars.

**NV-BART** (Xsum) China is opening up its roads to self-driving car tests on public roads and under certain conditions.

---

**CNN/DailyMail**

---

**Article** (CNN)Share, and your gift will be multiplied. That may sound like an esoteric adage, but when Zully Broussard selflessly decided to give one of her kidneys to a stranger, her generosity paired up with big data. It resulted in six patients receiving transplants. That surprised and wowed her. "I thought I was going to help this one person who I don't know, but the fact that so many people can have a life extension, that's pretty big," Broussard told CNN affiliate KGO. She may feel guided in her generosity by a higher power. "Thanks for all the support and prayers," a comment on a Facebook page in her name read. "I know this entire journey is much bigger than all of us. I also know I'm just the messenger." CNN cannot verify the authenticity of the page. ... (Beyond 1024 tokens input)

**Summary** Zully Broussard decided to give a kidney to a stranger.

**BART** (CNN/DM) Zully Broussard gave one of her kidneys to a stranger. Her generosity paired up with big data. It resulted in six patients receiving transplants. "I thought I was going to help this one person who I don't know," she says. The chain of surgeries is to be wrapped up Friday.

**NV-BART** (CNN/DM) Zully Broussard gave one of her kidneys to a stranger. Her generosity paired up with big data. It resulted in six patients receiving transplants. "I thought I was going to help this one person who I don't know, but the fact that so many people can have a life extension, that's pretty big"

**BART** (Xsum) A San Francisco woman has become the first person in the world to donate a kidney to six people at once.

**NV-BART** (Xsum) A woman in California has become the first person in the world to donate a kidney to a stranger in a "super-donor chain" by giving one of her own organs to another person in exchange for another person's kidney, in what is being hailed as an act of generosity.

---

**WikiHow**

**Article** Adult humans should drink 2-3 litres(or roughly eight 8 oz glasses) of water per day whilst children should drink 1-2 litres ( or roughly five 8 oz glasses.That is in addition to things like tea and coffee. Water keeps bodies at the correct temperature and removes toxins that are the inevitable result of metabolism and industrial life. You'll be healthier automatically.
**Summary** Drink more water.

**BART** (CNN/DM) Adult humans should drink 2-3 litres(or roughly eight 8 oz glasses) of water per day. Water keeps bodies at the correct temperature and removes toxins. Water also clears your skin, helps your kidneys, helps to control your appetite, and keeps you energized. It also keeps you from drinking unhealthy beverages like soda and juice.
**NV-BART** (CNN/DM) Adult humans should drink 2-3 litres of water per day. Children should drink 1-2 litres ( roughly five 8 oz glasses) Water keeps bodies at the correct temperature and removes toxins. Water also clears your skin, helps your kidneys, and keeps you energized. It also keeps you from drinking unhealthy beverages like soda and juice.
**BART** (Xsum) If you want to be healthier, you need to drink more water.
**NV-BART** (Xsum) If you want to be healthier, and have better skin, you should drink more water.

