# OpenReview forum: "Nonparametric Variational Regularisation of Pretrained Transformers"
_colmweb.org/COLM/2024/Conference — COLM_

### Official Review · Reviewer_nwsJ · 2024-05-08

**Rating:** 4
**Confidence:** 4
**Ethics Flag:** 1

**Summary:**

This manuscript proposed to leverage the idea of nonparametric variational information bottleneck method that modify the original attention module to perform post-training regularization on the pre-trained transformer model. The authors demonstrate the effectiveness on keeping the identity with specific initialization and out-of-domain generalization with additional regularization on some transformer architectures.

**Questions To Authors:**

* Could you revise Section 3 to make the prior distribution, likelihood and posterior distribution more clear? Specifically, I think it is necessary to show them in some equations, instead of some descriptive texts.
* What’s the meaning of $Z^2$ given $Z$ is a vector (e.g. in Equation 7)?
* Could you provide an algorithm box showing what’s your modification compared with the standard attention mechanism?
* What’s the motivation to set the parameters as in Equation 7? I don’t get any motivation on the necessity of $Z^2$.

**Reasons To Accept:**

* The idea is interesting and elegant.

**Reasons To Reject:**

* The manuscript is extremely hard to follow, even for a Bayesian expert.
* There are several points not clear to me even after I go through the manuscript for many times. See the Questions part below.

---

> ### Author Rebuttal · Authors · 2024-05-29
>
> We thank Reviewer **nwsJ** for their clarifying questions. We are pleased they found our paper **interesting and elegant**.
>
> *The manuscript is extremely hard to follow*
>
> We appreciate that the presentation of the relevant aspects of prior work is very compressed, and we agree that we need to make this explanation more accessible in any future version.  This will make the Bayesian nature of our model clearer, in addition to the answers below.
>
> *Make the prior distribution, likelihood and posterior distribution more clear?*
>
> We had hoped that the contribution of this paper could be understood without this explanation, but we are happy to add it as it clarifies the Bayesian nature of our contribution.  In section 3, we are inferring DP representations (distributions over mixtures) given the input.  The prior is a DP with a pseudocount $\alpha^p_0$ and mono-modal base distribution $G^p_0 = N(\mu^p,\sigma^p),$ which are set from the empirical distribution.  The NN converts the input into a set of pseudo-observations $(\mu^q,\sigma^q,\alpha^q)$.  Using exact inference of DPs, each of these pseudo-observations $(\mu^q_i,\sigma^q_i,\alpha^q_i)$ becomes one component $N(\mu^q_i,\sigma^q_i)$ of the posterior's base distribution $G^q_0$, which are added to the one component $N(\mu^p,\sigma^p)$ coming from the prior.  These components are weighted by their respective alphas (bottom pg3).  The likelihood is only used to justify the KL-divergence regulariser via the ELBo.  We will add a more explicit specification with precise equations to the paper.
>
> *What’s the meaning of $Z^2$ in Equation 7?*
>
> This and other algebraic computations are meant to be interpreted component-wise.  We will make this explicit.
>
> *Could you provide an algorithm box showing your modification to attention?*
>
> Yes. An algorithm or pseudocode box is a great idea. We can provide this in the appendix.
>
> *What’s the motivation to set the parameters as in Equation 7?*
>
> Intuitively, $\alpha_i$ determines the weight of the component for $z_i$, which we can see in equation (1) needs to be a function of the $L^2_2$ norm of $z_i$.  For this reason we allow a log-quadratic relationship between $z_i$ and $\alpha_i$ in equation (7).  We can make this explanation in the paper clearer.  More precisely, the bias term in equation (4) needs to go to a constant to get equivalence, which again requires a log-quadratic relationship between $\alpha_i$ and $z_i$ given the linear relationship between $z_i$ and $\mu_i$.

---

> > ### Comment · Reviewer_nwsJ · 2024-06-03
> > **Thanks for your clarification.**
> >
> > Thanks for your clarification. Indeed I think this manuscript can be interesting and elegant with some potential impacts. However the overall writing needs to be improved to make sure the core component can be well understood by the general audience. I would rather encourage the authors to polish the paper and re-submit.

---

### Official Review · Reviewer_nToS · 2024-05-10

**Rating:** 6
**Confidence:** 3
**Ethics Flag:** 1

**Summary:**

The authors present an initialization for a Bayesian Nonparametric Variational Information Bottleneck (NVIB) interpretation of transformer models, and demonstrate that there exists an identity initialization of the NVIB hyper-parameters such that NVIB attention is equivalent to standard attention. They also show that by tuning the NVIB hyper-parameters away from this identity initialization using priors derived via forward passes through the training data they can produce models with equivalent performance but greater uncertainty. Such tuning yields performance gains for a BART summarization model trained on XSum, but does not yield significant gains for a BART summarization model trained on CNN/Daily Mail nor a Marian translation model trained on OPUS100.

Minor issues:
* "HuggingFace Wolf et al. (2020)." => "HuggingFace (Wolf et al. 2020)."
* "BART Lewis et al. (2020)" => "BART (Lewis et al. 2020)"

**Questions To Authors:**

* Is there prior research that would have suggested that quantization would improve performance? That wouldn't have been my intuition. I'm trying to understand the decision to use quantization as a baseline in this paper. The results certainly show that in the datasets selected quantization does not improve performance.

**Reasons To Accept:**

* Showing that NVIB attention can be initialized such that it matches standard attention seems like a useful result.
* The post-training regularization approach allows model uncertainty to be increased without decreasing performance and without back propagation or parameter updates.
* Summarization and translation models are evaluated with the proposed regularization both in domain and out of domain

**Reasons To Reject:**

* The paper is more theoretically interesting than practically useful, as the regularization doesn't generally yield benefits, even in out-of-domain settings. (Gains are only observed in the specific combination of BART trained on XSum; in all other settings performance is similar to the original model.)
* The paper is hard to read in several places because it offloads too much to the appendix, starting a discussion of an analysis but then finishing the explanation of that analysis and the figures showing the results in the appendix.

---

> ### Author Rebuttal · Authors · 2024-05-29
>
> We appreciate Reviewer **nToS** for their analytical view of our experiments section. We are pleased they emphasized the contributions of: our **useful** identity initialisation, which allows access to pretrained models; Post-training regularisation **without decreasing performance** and  **without backpropagation**; validating our results on **summarization** and **translation** across **in domain** and  **out of domain**.
>
> *The regularization doesn't generally yield benefits, even in out-of-domain settings.*
>
> We appreciate that Reviewer **nToS** found our paper **theoretically interesting**.  We believe that the empirical results are sufficient to support these theoretical claims about the nature of pretrained transformers, given that there are small improvements in most cases. In general, regularisation does not guarantee improved performance.  It is surprising there exists any improvements that generalise to test set performance given that this method is applied post-training without backpropagation or parameter updates. We believe that the variance in performance is due to the nature of the different tasks. Summarisation is fundamentally an information compression task, whereas translation is a less compressive transformation between languages (mentioned top pg9). We conjecture that the larger improvement in NV-BART(Xsum) is due to the abstractive nature of the Xsum dataset (mentioned mid pg8). This requires more generalisation than the extractive CNN/DM dataset which relies on selecting a summary from the document. Some discussion of this issue is already in the paper, but we will add further discussion if space allows.
>
> *The paper is hard to read in several places because it offloads too much to the appendix*
>
> We agree that the analysis paragraph in section 4.2 would be much improved by moving some representative plots from the appendix into the body of the paper, and we will do this for any camera-ready version.  This was necessary given space constraints.  We will expand the discussion in the body of the paper, as space allows, but given the content quantity, the appendix will still be necessary.
>
> *Is there prior research that would have suggested that quantization would improve performance?*
>
> Yes. "We find that reducing the precision steadily from 16 to 4 bits increases the zero-shot performance" - The case for 4-bit precision: k-bit Inference Scaling Laws - Dettmers - ICML2023

---

### Official Review · Reviewer_SbeV · 2024-05-12

**Rating:** 6
**Confidence:** 3
**Ethics Flag:** 1

**Summary:**

This paper explores a method based on NVIB (Henderson and Fehr, ICLR 2023) to apply as regularization over multi-head transformers for OOD text summarization and machine translation tasks.

In an encoder-decoder model, BART, the NVIB projection layer combines the Dirichlet Process-based empirical prior with the transformer embeddings to produce a representation for denoising MHA. In the NV-Marian case, the improvement of NVIB is less significant.

I spent some time reading NVIB in two of its original references; it appears to be a relatively new and preliminary direction with only a few studies.

From my perspective, it is acceptable if NVIB does not serve as the best regularization/post-training boosting method for some applications. For example, motivating and explaining the limitations of this form of non-parametric variational Bayesian application in language modeling would be interesting to general audiences.

If the authors could add some studies on a decoder-only language model with NVIB (e.g., small-scale GPT-2 or LLama), it could potentially increase the general impact and interest of this work across more language modeling applications.

I think it is borderline paper. I enjoy the general writing and presentation of this work.

**Questions To Authors:**

1. Would it be beneficial to add some min-max perturbation to simulate severe OOD cases for further diving into the properties of NV-based LM? One intuition is that the current performance is relatively diverse and seems to perform better in OOD tasks.


2. This might be a bit open. There are some causal variational Bayesian works [A,B] in robust RL and sequence modeling that improve OOD robustness via local Lipschitz constant measurement. Could adding a causal graph over the VAE framework further help with some counterfactual OOD evaluation cases?

***

A. Large Language Models and Causal Inference in Collaboration: A Comprehensive Survey

B. Training a Resilient Q-Network against Observational Interference

C. Lipschitz Parametrization of Probabilistic Graphical Models

**Reasons To Accept:**

1. The work aims to extend the nonparametric mixture distributions of transformer to applying regularization post-training.

2. On OOD sub-tasks of summarisation, the performance NV-BART is significantly better than post-training methods.

**Reasons To Reject:**

1. the performance of NV-based LM seems to be little bit inconsistent and dataset dependent. Some in-depth analysis is encouraged.

---

> ### Author Rebuttal · Authors · 2024-05-29
>
> We would like to thank Reviewer **SbeV** for their thorough review, and for taking the time to read the prior work, which clearly helped in understanding our contribution. We are delighted they: **enjoy the general writing and presentation**; emphasized the novelty of our **new and preliminary direction** and highlighted our potential by performing **significantly better than post-training methods**.
>
> *If the authors could add some studies on a decoder-only language model with NVIB*
>
> We certainly agree that leveraging the submitted work to apply NVIB to pretrained LLMs would improve the impact of this work, but this is a complex topic which cannot be adequately addressed in the length of this paper.  This topic is part of our current work, and we would be happy to include it in an extended discussion of future work.
>
> *The performance of NV-based LM seems to be little bit inconsistent and dataset dependent. Some in-depth analysis is encouraged.*
>
> We consider any improvement as an achievement for a method which is applied post-training without backpropagation or parameter updates, as shown by the comparison to baselines.
> Secondly, we believe that the variance in performance does not reflect inconsistent performance of NVIB but rather is due to the nature of the different tasks. Summarisation is fundamentally an information compression task, whereas translation is a less compressive transformation between languages (mentioned top pg9). We speculate the larger improvement in NV-BART(Xsum) is due to the abstractive nature of the Xsum dataset (mentioned mid pg8). This requires more generalisation than the extractive CNN/DM dataset which relies on selecting a summary from the document. In general, regularisation does not guarantee improved performance. Some discussion of this issue is already in the paper, but we will add further discussion if space allows.
>
> *add some min-max perturbation to simulate severe OOD cases.*
>
> This is a very interesting and informative suggestion, and we will definitely investigate how we can construct such an evaluation.
>
> *Could adding a causal graph over the VAE framework further help with some counterfactual OOD evaluation cases?*
>
> This suggestion and the cited papers look very interesting, but it will require further study.  Causal graphs and Counterfactual OOD are very relevant, and we think this could be a strength of our Bayesian approach, but it is out of the scope for the current paper.

---

> > ### Comment · Reviewer_SbeV · 2024-06-06
> > **Re: Rebuttal**
> >
> > Thank the authors for the response.
> >
> > Given the current scope is still fixed (e.g., decoder lm exclusive & evaluation on OOD), I will keep my current score. If the authors will revise the presentation on mentioning the challenges and related directions discussed here, the final version would be with boarder impacts and audiences.
> >
> > It is a borderline and interesting paper to me but I won’t strongly argue for acceptance.

---

### Official Review · Reviewer_59aH · 2024-05-22

**Rating:** 6
**Confidence:** 3
**Ethics Flag:** 1

**Summary:**

The paper extends NVIB regularization to all forms of multi-head attention in a transformer model. Also, it shows that existing pretrained transformers can be reinterpreted as nonparametric variational models using an empirical prior distribution and identity initialization. Experimental results show the effectiveness of the proposed approach.

**Questions To Authors:**

- When used during training, the NVIB layer provides an information-theoretic, sparsity-inducing regulariser over attention-based representations. Is there any advantage over just keeping top-k weights, though we have to set k in advance?

- For denoising attention, how should we understand the concept of denoising? For example, in diffusion generative modeling, we want to recover clean images from noisy ones, involving a denoising process. How does denoising work in your context?

- Is it possible to link this research to LLMs?

**Reasons To Accept:**

Model understanding can benefit the community. Overall, I think the paper is solid in some sense, but needs modification. It seems to be a new research direction without too many studies.

-------

Update after rebuttal: I raised the score --> 6.

**Reasons To Reject:**

- Writing issue. The paper starts with the NVIB, however, without sufficient background, section 2 is a bit hard to follow.

---

> ### Author Rebuttal · Authors · 2024-05-29
>
> We would like to thank Reviewer **59aH** for their time, effort and clarifying questions. Furthermore, we are pleased they highlighted the strengths of our contribution: **model understanding that can benefit the community**; Novelty of our **new research direction**; and rigour by being **solid**.
>
> *Writing issue. The paper starts with the NVIB, however, without sufficient background, section 2 is a bit hard to follow.*
>
> We appreciate that section 2 is a very compressed presentation of the relevant aspects of prior work, and we agree that we need to use the extra page available for any camera-ready version to make this explanation more accessible.  However, for scientific diversity and developing new research directions, it must be possible to build on top of previously published work without reiteration. We believe that section 2 adequately introduces the concepts which are needed to understand our contribution.
>
> *Advantage over just keeping top-k weights?*
>
> This question does not directly apply to the submitted paper because we evaluate NVIB entirely post-training.  For training, NVIB encourages a continuous transition from not-sparse to sparse models, whereas top-k pruning of vectors introduces a discrete discontinuity between not-pruning and pruning vectors, which is potentially an advantage for NVIB, as well as its strong information-theoretic foundations.
>
> *Understand the concept of denoising?*
>
> In this paper, denoising attention just uses Bayesian denoising as a mathematical function used to define the generalisation of attention.  In the previous work, Nonparametric VAEs (like all VAEs) are closely related to diffusion in that they introduce noise to the embeddings during training, which the model needs to learn to remove to reconstruct the input.
>
> *Is it possible to link this research to LLMs?*
>
> Certainly; this is a topic we are currently working on, and we would be happy to discuss it more under future work. However, there are a number of complexities in LLMs (e.g. relative position embeddings, flash-attention and scaling for compute) which make this work outside the scope of the submitted paper.  The submitted paper is an essential step towards NVIB for LLMs. As LLMs get larger, the value of regularisation should increase.  Also, during LLM generation, NVIB would allow sampling to be done for the embeddings of the model, resulting in more diversity in generation.

---

> > ### Comment · Reviewer_59aH · 2024-06-04
> > **Update after rebuttal**
> >
> > Sorry for the late reply and thank you for your clarification.
> >
> > I think the paper could be sufficiently clarified given an extra page in the camera-ready version. I raised my score.

---

### Decision · Program_Chairs · 2024-07-10

**Decision:**

Accept

**Comment:**

Henderson & Fehr (2023) offer a new view of the attention component of transformers through Neural Variational Information Bottleneck. This paper takes the idea further, extending from single-head cross-attention to all forms of attention in the transformer. The reviewers in general find the ideas to be sound and mathematically elegant. Personally I am happy to see work that explores new foundations for the modeling components of LLMs, and would be interested to learn more about the ideas presented in the paper.

However, as a line of research, NVIB regularization is still somewhat work-in-progress, particularly from the empirical perspective. The reviewers did not find the empirical results to be very persuasive: I agree with the reviewer suggestion to consider more extreme distribution shift; and, given the well-known problems with Rouge- and Bleu-based evals, I wonder whether it would be possible to apply prompted evals, particularly for the summarization task.

Another area for improvement is in clarity of presentation. The reviewers, some of whom have quite strong mathematical backgrounds, also found the presentation to be difficult to follow, and make useful suggestions for improvement that would likely benefit many CoLM readers. Reviewer `nwsJ` had particularly concrete ideas for improvement, which I endorse. I also found it unfortunate that most of the novel aspects of the work had to be relegated to appendix B.

[comments from the PCs] Please follow up on the AC and reviewer recommendations to improve the presentation. Improving the presentation will help your work have bigger impact.